# Synthetic microbe-to-plant communication channels

Alice Boo [1,3], Tyler Toth[1,3], Qiguo Yu[1], Alexander Pfotenhauer[2], Brandon D. Fields[1], Scott C. Lenaghan [2], C. Neal Stewart Jr [2] & Christopher A. Voigt [1] ✉

Plants and microbes communicate to collaborate to stop pests, scavenge nutrients, and react to environmental change. Microbiota consisting of thousands of species interact with each other and plants using a large chemical language that is interpreted by complex regulatory networks. In this work, we develop modular interkingdom communication channels, enabling bacteria to convey environmental stimuli to plants. We introduce a "sender device" in *Pseudomonas putida* and *Klebsiella pneumoniae*, that produces the small molecule *p*-coumaroyl-homoserine lactone (pC-HSL) when the output of a sensor or circuit turns on. This molecule triggers a "receiver device" in the plant to activate gene expression. We validate this system in *Arabidopsis thaliana* and *Solanum tuberosum* (potato) grown hydroponically and in soil, demonstrating its modularity by swapping bacteria that process different stimuli, including IPTG, aTc and arsenic. Programmable communication channels between bacteria and plants will enable microbial sentinels to transmit information to crops and provide the building blocks for designing artificial consortia.

Communication between plants and microbes consists of a rich language of chemical messages[1–4]. Plants release 100s of molecules from their roots, using up to 50% of the plant's photosynthetic output to communicate with thousands of bacteria and fungi[5–7]. The exudate composition spans small volatile organic acids, flavonoids, lipids, oligosaccharides, peptides, and proteins that are received by bacteria that, in turn, can respond to the plant with their own chemical signals[2,6,8–10]. Manipulating plant-microbe communication has been used for crop improvement; however, system complexity limits predictability[1,8,11–15]. In engineering projects, defined communication channels would facilitate the distribution of functions across an agriculture system. For example, bacterial sentinels could survey the soil using genetically-encoded sensors and circuits and transmit the information to the plant.

Synthetic biology projects often harness communication to coordinate cells in space and time[16,17]. A channel consists of a genetically-encoded "sender device" that produces a diffusible small molecule and "receiver device" that responds to it[18–20]. The term "device" refers to a transcriptional signal serving as the input (sender) or output (receiver), which simplifies the connection to other devices to build a larger system. Commonly, the chemical signals are acyl-homoserine lactones (acyl-HSLs) gleaned from bacterial quorum sensing systems. The acyl-HSL is produced by a single enzyme and binds to a regulatory protein. Specificity is determined by the length of the acyl chain, which has been exploited to build multiple non-interfering channels[21–24]. This language has been used for a plethora of projects, including stabilizing biofilm consortia, distributed computing, and timing metabolic flux in a bioreactor[16,25–34]. Communication channels between eukaryotes have been developed based on peptides and pheromones[35–37].

Synthetic plant-to-microbe communication channels have been developed[38]. Plants can be engineered to excrete new chemicals from their roots, the receiver for which is put in a bacterium[39,40]. For example, tobacco was engineered to produce acyl-HSL and this could induce *Escherichia coli* carrying an acyl-HSL receiver[41]. Similarly, a

[1]Department of Biological Engineering, Synthetic Biology Center, Massachusetts Institute of Technology, Cambridge, MA 02139, USA. [2]Center for Agricultural Synthetic Biology, University of Tennessee, Knoxville, TN 37996, USA. [3]These authors contributed equally: Alice Boo, Tyler Toth. ✉e-mail: cavoigt@gmail.com

sender device in barley was built by introducing two prokaryotic genes to make scyllo-inosamine (SI), the receiver for which is the SI-binding MocB regulator, placed in in the soil bacterium *Azorhizobium caulinodans*[38,42].

Microbe-to-plant communication requires building a sensor in the plant with a low limit-of-detection. Constructing sensors in plants is difficult due to slow engineering cycles, tissue-specific expression, fewer genetic part (e.g., promoter) libraries, chromosome context effects and complex molecular transport[43–48]. Genetically-encoded plant sensors have been built that respond to ethanol, tetracycline, steroids, insecticides, trinitrotoluene (TNT), copper, fentanyl, and acetaldehyde[44,49–58]. However, these are not appropriate communication signals because of issues with specificity, diffusion, high limits-of-detection, or low production titers by bacteria.

Because of their role in establishing symbiotic relationships between bacteria and plants, acyl-HSLs have been proposed to be natural examples of interkingdom communication[2,59,60]. Biofilms on the root are abundant in acyl-HSLs, reflecting the volume of communication that occurs there[59,61–64]. Indeed, up to 12% of the species in soil make acyl-HSLs[59,64,65]. Plants have evolved means to eavesdrop on them to identify bacteria and respond appropriately[1,2]. Acyl-HSLs rapidly diffuse to the root surface from up to 30 μm away, are taken up and regulate hundreds of genes via poorly understood mechanisms[1,66–72]. The specific response depends on the plant, but short-chain acyl-HSL tend to change root morphology whereas long-chain (>C12) acyl-HSLs affect defense and immunity, and both can impact energy/metabolic process, hormone production and $Ca^{2+}$ signaling[70,71,73,74]. Plants and microbes can interfere with acyl-AHSL signaling by producing degrading enzymes and chemical mimics[66,70,74,75].

*Rhodopseudomonas palustris* is a plant-growth promoting bacterium isolated from rice paddies that produces only one quorum signal: *p*-coumaroyl-homoserine lactone (pC-HSL)[76]. In place of the acyl-group, it sources an aryl-group from *p*-coumarate secreted from plant roots[65,77]. No known soil bacterium has the complete pathway to pC-HSL[65]; however, a synthetic pathway (*rpaI/4cl/tal*) has been built in *E. coli* to make *p*-coumarate and incorporate it into pC-HSL[78,79]. The activator RpaR binds to pC-HSL and to the *rpaO*A* DNA operator[65,78]. Some other species make pC-HSL, but it is far less abundant than acyl-HSL[59,64,65], plants do not respond except at high concentrations[80,81], and there are far fewer deactivating enzymes and mimics in soil[61,82].

Here, we demonstrate programmable microbe-to-plant communication from a pC-HSL sender in the soil bacteria *Pseudomonas putida* KT2440 and *Klebsiella pneumoniae* 342 to pC-HSL receivers in *Arabidopsis thaliana* and *Solanum tuberosum* (potato) (Fig. 1a). *P. putida* has been proposed to be used in agriculture to promote plant growth, is non-pathogenic and does not secrete any HSLs[83–85]. *Klebsiella pneumoniae* 342 is an endophytic nitrogen fixer first isolated from maize, colonizes roots of maize, wheat, rice, and Arabidopsis, and does not secrete any HSLs[86–89]. No plant pC-HSL receptor was known, so here we build a receiver device for plants by constitutively expressing RpaR and building a responsive promoter. This receiver is specific to pC-HSL and does not cross-react with acyl-HSL. The sender device can be connected to different sensors (IPTG, aTc, arsenic) and logic circuits and communicate the output to the plant root (Fig. 1b). This approach allows the plant to respond to different environmental signals by swapping the bacterium, rather than genetically modifying the plant. This work describes the division of labor by moving environmental sensing to bacterial sentinels at the roots, who relay the information to the plant.

## Results
### Plant HSL receiver devices
The plant receiver should detect a specific HSL and respond by generating a transcriptional output; in other words, activating a promoter (Fig. 1c). The receiver design was based on a plant promoter scaffold developed by Quatrano and co-workers that was shown to be functional in different plants, including *A. thaliana*[90]. This promoter was based on a minimal 35S motif ($P_{m35S}$) that is inactive unless an activator binds upstream. This scaffold had been used to create a 3-oxooctanyl-L-HSL (OC8-HSL) inducible promoter that cross-reacts with other HSLs[90]. Here, we changed this design to decrease the spacing between four operators from 10 bp to 2 bp, following our work in mammalian cells[91], and added the TMVΩ translational enhancer[92] downstream of the promoter to increase expression.

Four prokaryotic regulators were selected from quorum sensing systems: 1. LuxR^AM (*Vibrio fischeri*), 2. CinR^AM2 (*Rhizobium leguminosarum*), 3. LasR^AM (*Pseudomonas aeruginosa*) and 4. RpaR^AM (*Rhodopseudomonas palustris*). Each regulator and its cognate operator were gleaned from a set that we had previously evolved to improve their orthogonality and dynamic range (indicated by ^AM)[21,93]. To make the regulators functional in plants, their N-termini were fused to a SV40 nuclear localization signal (NLS), activation domain (VP16) and a 6 G flexible linker (Fig. 1d)[94]. The regulators were placed under the control of the strong 35S promoter, TMVΩ translation enhancer, and $T_{OCS}$ terminator.

The devices were constructed by combining the reporter expression cassette with the output promoter (Methods). The regulators are constitutively expressed as monomers and, in the presence of the HSL, they dimerize and bind to the output promoter, leading to green fluorescent protein (GFP) expression. The cassette included a phosphinothricin acetyltransferase gene to generate resistance to the herbicide phosphinothricin (PPT) as a selectable marker. The constructs were transformed into *A. thaliana* using the *Agrobacterium* floral dip method (Methods). Multiple homozygous *A. thaliana* lines for each of the receivers were identified after rounds of herbicide selection that segregated to all resistant progenies. No significant phenotypic differences were observed between wild-type *A. thaliana* and *A. thaliana* containing the pC-HSL receiver (315_14_5) (Fig. 1e, f).

The *A. thaliana* lines containing the receivers were then tested for their ability to respond to their cognate HSLs. Seeds were germinated and grown on agar plates for 7–12 days in a growth chamber before being transferred to a hydroponic system, following the protocol of Shank and co-workers (Methods) (Supplementary Fig. 1)[95]. This system allowed the plant roots to be exposed to a homogeneous inducer concentration in a 24-well plate format. We screened 4–9 plant lines for each HSL receiver. As an initial screen of activity, we added 100 μM of HSL inducer: N-3-oxohexanoyl-L-homoserine lactone (OC6-HSL) for the LuxR-expressing line, 3-hydroxytetradecanoyl-homoserine lactone (OHC14-HSL) for the CinR-expressing line, N-3-oxododecanoyl-L-homoserine lactone (OC12-HSL) for the LasR-expressing line or pC-HSL for the RpaR-expressing line. After 24 h of induction in the hydroponic plate, the fluorescence was visualized in the root tissue using confocal microscopy (Methods). To quantify fluorescence, we calculated the mean pixel intensity (MPI) across root tissue sections (Supplementary Fig. 2).

*A. thaliana* lines containing the pC-HSL, OC12-HSL and OHC14-HSL receivers showed 180-fold, 40-fold and 7-fold inductions, respectively (Fig. 2a, b and Supplementary Figs. 3–5). The OC6-HSL receiver yielded no functional lines, so it was not pursued further. Of the nine independent lines tested for pC-HSL induction, seven were active, of which we selected *A. thaliana* 315_14_5 for further characterization (Supplementary Fig. 6). We found that GFP was only expressed in mature tissues, with no GFP in the meristem (Fig. 2a, Supplementary Figs. 7–10). Using RT-qPCR, we found that GFP was mostly expressed in root tissues compared to leaf and stem tissues (Supplementary Fig. 11).

The full response functions were measured for the pC-HSL and OC12-HSL receivers (Fig. 2b and Supplementary Fig. 3). The minimum detection limit was 100 nM pC-HSL, which is an order of magnitude

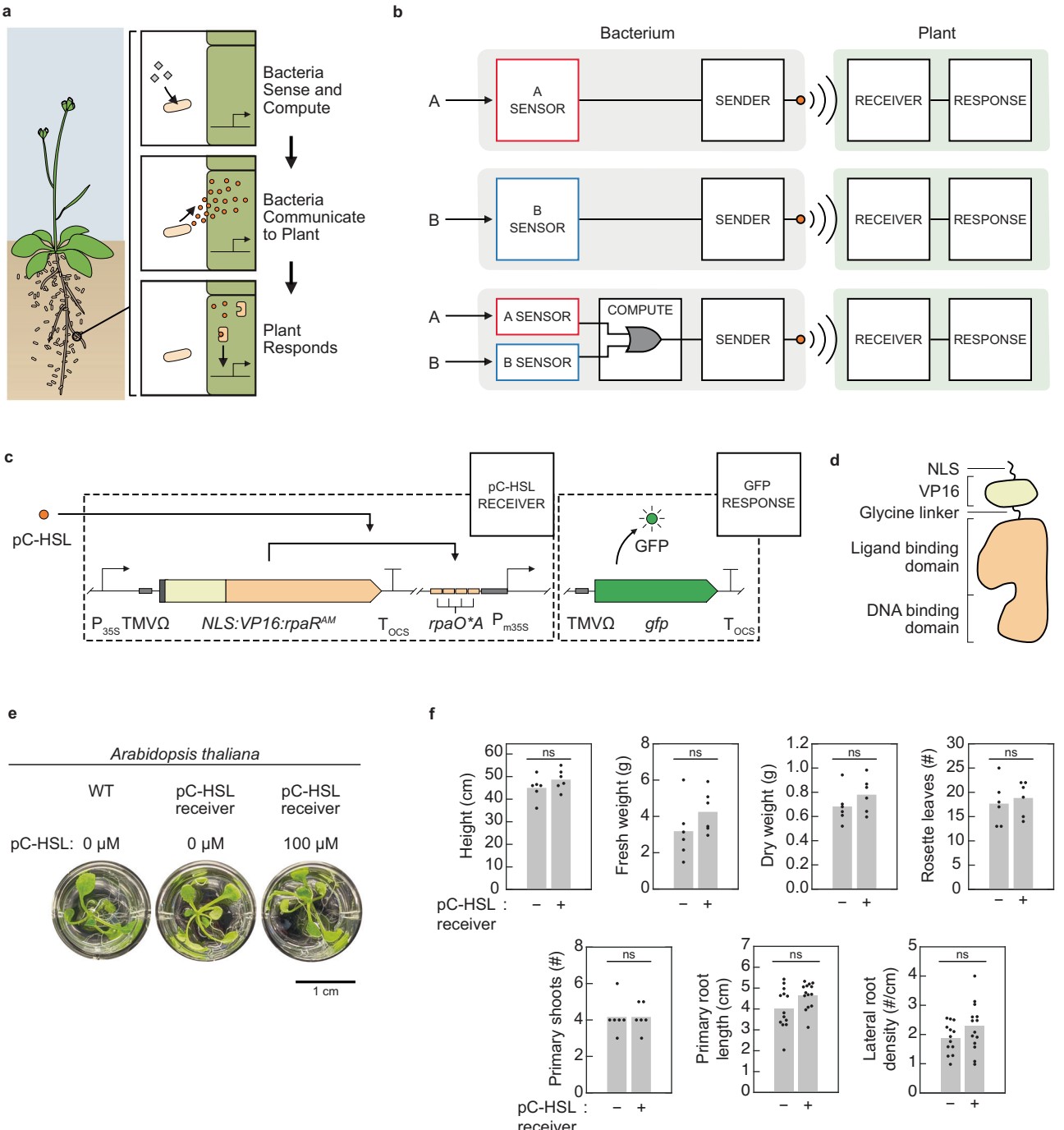

**Fig. 1 | Design of bacteria-to-plant communication. a** Bacteria receive a signal in the soil (grey diamonds) that induces the release of the communication signal (orange circles) to be sensed by regulatory proteins in the plant cell. **b** The communication channel is modular. To change the signals to which the plant responds, it simply can be grown with different bacteria engineered to connect different sensors (A or B) to the synthesis of the chemical used for communication. The bacterium can also integrate these signals using genetic circuits; an OR gate is shown. **c** The plant pC-HSL receiver device. The genetic part DNA sequences are provided in Supplementary Data 1. **d** The modifications to the prokaryotic RpaR regulator (orange) are shown to make it functional in plants. **e** Phenotypic comparison of *A. thaliana* wild-type to that carrying the pC-HSL receiver (*A. thaliana* 315_14_5). The plants were induced for 24 h in MS media in the hydroponic system. **f** Phenotypic comparison of wild-type *A. thaliana* with that carrying the pC-HSL receiver (*A. thaliana* 315_14_5_1) grown in soil. The data points represent replicates performed with different plants ($n = 6$ for height, fresh weight, dry weight, number of rosette leaves, number of primary shoots and primary root length; $n = 13–14$ for primary root length and lateral root density) and the bars represent the means of these points. Statistically significant differences were determined using two-tailed Student's $t$ test (ns, not significant $P > 0.05$). Source data are provided as a Source Data file.

higher than the detection limit of RpaR in *R. palustris*[65]. In the absence of inducer, the background expression was 5-fold higher than wild-type *A. thaliana* (Supplementary Fig. 13). The whole root response functions for the pC-HSL and OC12-HSL receivers are shown in Supplementary Fig. 14.

Plants containing the pC-HSL receiver were then tested whether they respond to non-cognate HSLs (orthogonality) (Fig. 2c). Plants were grown and induced, as before, with 100 μM of each HSL for 24 h. There was no observed induction by the non-cognate HSLs, as the background fluorescence was indistinguishable from plants in the

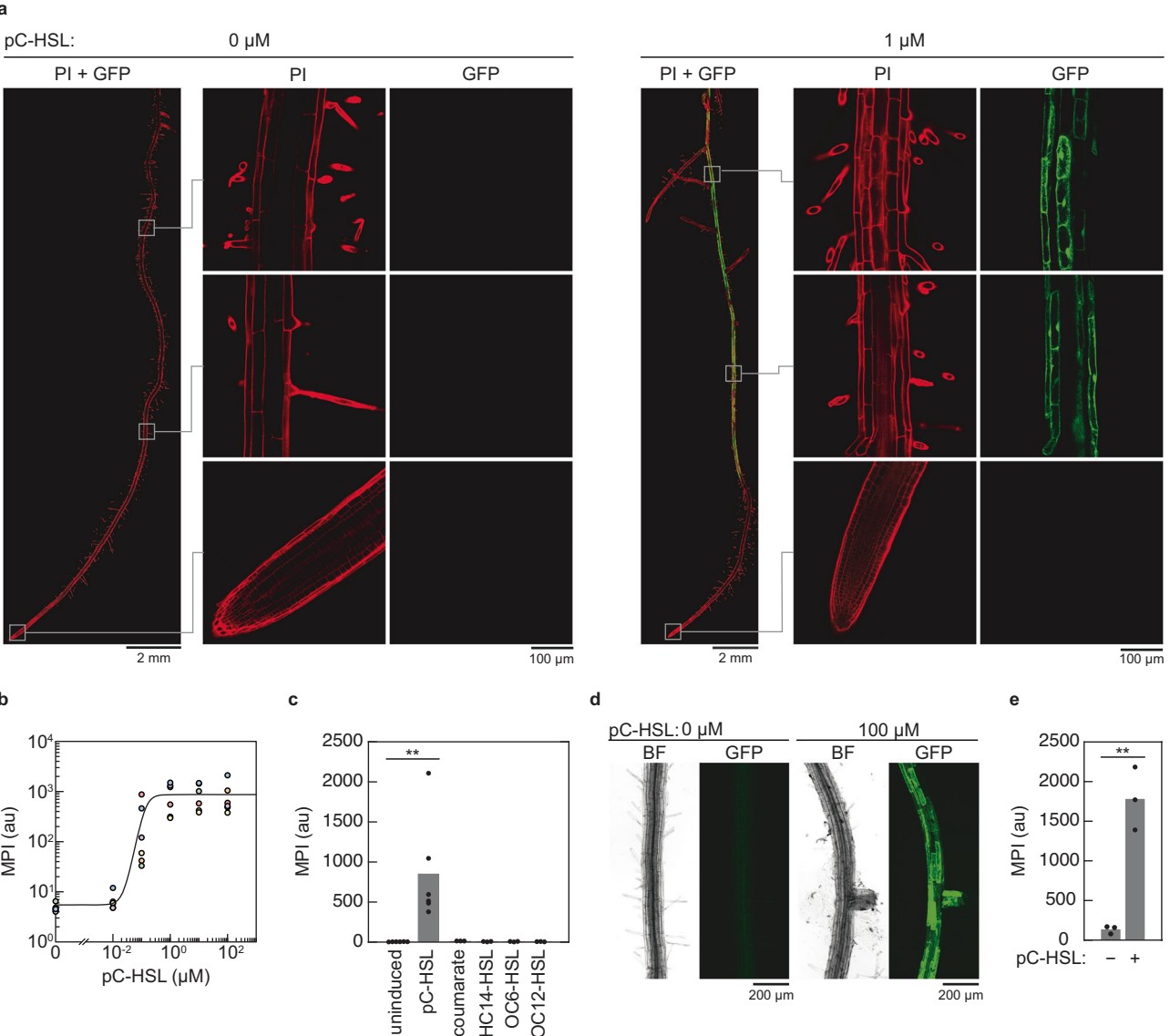

**Fig. 2 | The _A. thaliana_ pC-HSL receiver. a** Fluorescence microscopy images of the induction of the pC-HSL receiver expressing GFP (green) and stained with propidium iodide (PI, red). _A. thaliana_ 315_14_5_1 was induced with 1 μM pC-HSL for 24 h in a hydroponic system (Methods). Images are representative of experiments performed on three different days with different plants (Supplementary Fig. 7). **b** Response function of the _A. thaliana_ pC-HSL receiver. Each color represents experiments repeated on 6 different days with different plants (_A. thaliana_ 315_14_5). All the data were fit to Eq. 1 (parameters in Supplementary Table 1). Raw images used to calculate the MPI are provided in Supplementary Fig. 12. **c** Orthogonality of the pC-HSL receiver. _A. thaliana_ 315_14_5 was induced with 100 μM of each inducer (_p_-coumarate, OHC14-HSL, OC6-HSL, OC12-HSL and pC-HSL) for 24 h in a hydroponic system (Methods). The points represent replicates performed with different plants (_A. thaliana_ 315_14_5) on different days (n = 6 for pC-HSL and

uninduced, n = 3 for other HSLs, and n = 2 for _p_-coumarate) and the bars represent the means of these points. **d** Microscopy images of the induction of the pC-HSL receiver in soil. _A. thaliana_ 315_14_5_1 was grown and induced by watering the plants with 100 μM pC-HSL in sterile soil (Methods). Images are representative of experiments performed on three different days with different plants. **e** Induction of the _A. thaliana_ pC-HSL receiver in soil. The bars represent the mean fluorescence from three plants grown on different days (_A. thaliana_ 315_14_5). Raw images used to calculate the MPI are provided in Supplementary Fig. 16. There is a 13-fold upregulation between induction with 0 μM of pC-HSL and 100 μM of pC-HSL. Statistical significance was determined using two-tailed Student's _t_ test (***$P < 0.001$; **$P < 0.01$; *$P < 0.05$; ns, not significant $P > 0.05$). Source data are provided as a Source Data file.

absence of inducer. This was consistent with the observation that these molecules do not bind the evolved RpaR[AM21]. The precursor _p_-coumarate does not induce the receiver, which is consistent with previous studies with RpaR[96].

The inducibility of the pC-HSL receiver for plants grown in soil was tested (Fig. 2d, e). Seeds were sterilized and germinated on agar plates to achieve uniform germination. After the emergence of the first leaf, the plants were transferred into 1:3 vermiculite:soil non-sterile mix supplemented with fertilizers and grown for 10 days in a growth

chamber before being induced in situ, in the soil (Methods). Plants were induced by pipetting 1 mL of water supplemented with 100 μM of pC-HSL directly on the plant-soil interface. We estimated the concentration of pC-HSL throughout the soil to be 260 nM, but it is expected to be higher near the surface as it is added through watering (Methods). Plants were then grown for an additional 24 h in the growth chamber before being prepared for imaging by washing the roots in water. The fluorescence from GFP was measured using confocal microscopy (Methods). As observed in the hydroponics experiments,

GFP was only observed in the root tissue of plants containing the pC-HSL receiver in the presence of pC-HSL.

## Bacterial pC-HSL sender device

Soil bacteria often produce acyl-HSLs. For example, *P. putida* IsoF and WCS358 produce 3OC12-HSL[97–99]. To confirm that our strains do not produce acyl-HSLs or pC-HSL, we performed a BLAST search on the genomes of *P. putida* KT2440 and *K. pneumoniae* (Methods). Neither species had any proteins with significant sequence similarity to RpaI, LuxI, LasI, CinI, or TraI. The inability of these species to produce pC-HSL was further validated experimentally by testing whether the wild-type strains could induce the *A. thaliana* pC-HSL receiver (Supplementary Figs. 17–18).

A sender device must convert the transcriptional output of a sensor or circuit into the production of the communication molecule to a concentration detectable by the receiver. A three-gene operon was designed to convert endogenous tyrosine to pC-HSL (Fig. 3a). The biosynthetic pathway was constructed using a pC-HSL synthase gene from *Rhodopseudomonas palustris* (*rpaI*), a tyrosine ammonia-lyase gene from *Rhodobacter sphaeroides* (*tal*) and a 4-coumarate coenzyme A ligase gene from *Nicotiana tabacum* (*4cl*)[23,79]. For experiments requiring the production of pC-HSL, *p*-coumarate was added to the media because it increases pC-HSL production[79]. A red fluorescent protein gene (*mCherry*) was included in the operon so that induction could be monitored.

Initially, to test whether this pathway produced sufficient pC-HSL to theoretically turn on the plant receiver, the biosynthetic operon was placed under the control of the strong constitutive promoter BBa_J23100 on a pBBR1-ori plasmid. Before testing with plants, we built a surrogate receiver using *E. coli* engineered to respond to pC-HSL. The *E. coli* genome was modified to contain *rpaR*[AM] controlled by a constitutive promoter and $P_{rpaR*A}$ driving yellow fluorescent protein (YFP) expression[21,100] (*E. coli* MG1655 sTT658, Supplementary Table 4). The fluorescence of YFP was measured using flow cytometry (Methods). This reporter strain showed pM sensitivity and was fully induced at 10 nM pC-HSL after 3 h (Supplementary Fig. 19) (Methods). To estimate the pC-HSL concentration produced by sender cells, we collected the supernatant after growth in MS medium supplemented with 100 μM of *p*-coumarate for 24 h and used it to induce the *E. coli* pC-HSL receiver strain. Using these data, we estimated the concentration produced by the *P. putida* in the presence of *A. thaliana* to be $1.5 \pm 0.2\,\mu M$ pC-HSL (Supplementary Fig. 19). When *K. pneumoniae* was cultured in MS medium for 24 h in the presence of *A. thaliana*, pC-HSL production was estimated to be $0.30 \pm 0.07\,\mu M$.

## Bacteria-to-plant signal relay

We then tested the ability for bacteria producing pC-HSL to induce the *A. thaliana* pC-HSL receiver (Fig. 3a). Initially, the strains constitutively producing pC-HSL were used (*P. putida* pTT337 and *K. pneumoniae* pTT337). The plants were germinated on solid agar and added to the 24-well plates. Separately, the bacteria were grown in LB medium overnight, then washed and diluted into MS medium into the wells containing the plants. The co-culture was grown for 24 h, and the roots imaged (Fig. 3b). The *A. thaliana* pC-HSL receiver was induced 30-fold when cultured with *P. putida* pTT337 and 50-fold with *K. pneumoniae* pTT337 producing pC-HSL but remained uninduced when grown with wild-type bacteria (Fig. 3c, d, Supplementary Figs. 17–18, 20). Induction in solid agar produced a similar result as the hydroponic system (Supplementary Fig. 21).

To test for non-specific interference in the communication between the plant and the bacteria, we then tested whether the receiver's response to pC-HSL changed in the presence of wild-type *P. putida*, which could happen through a non-specific response to the bacteria. We did not observe a significant change in the response function when the bacteria were present (Supplementary Fig. 22).

The ability to transmit a signal in soil was then tested (Fig. 3e, f, Supplementary Fig. 23–24). Bacteria were introduced to the soil either by diluting them to $OD_{600} = 0.1$ into PBS supplemented with 100 μM *p*-coumarate and subsequently dispensing them into the soil, or by immersing the sterile *A. thaliana* pC-HSL receiver seeds into the *P. putida* overnight culture before sowing them into the soil (Methods). In sterile soil, we observed 32-fold and 39-fold inductions of the pC-HSL receiver when the system was inoculated with *P. putida* pTT337 compared to *P. putida* WT through seed inoculation and watering respectively (Fig. 3e). In non-sterile soil, a smaller 17-fold induction was observed. Whole-root imaging of the plants inoculated by watering in sterile and non-sterile soil show that GFP is expressed in the mature root tissues closer to the surface (Supplementary Fig. 24).

## Induction of potato by bacteria producing pC-HSL

The pC-HSL receiver device was then moved to potato (*S. tuberosum*). The pC-HSL receiver construct (pTT315-Hyg, Supplementary Fig. 34) was used to construct *S. tuberosum* 315 (Methods). Carrying the pC-HSL receiver resulted in a small decrease in both the fresh and dry stem weight, and a small increase in the chlorophyll content index, but otherwise did not have a significant impact on the phenotype (Supplementary Fig. 25). The response of the *S. tuberosum* pC-HSL receiver was measured by adding pC-HSL to the plants in the hydroponic system. After 24 h, RNA was isolated from the root tissue and *gfp* transcription was quantified by qRT-PCR (Supplementary Fig. 26). qRT-PCR was used instead of confocal microscopy due to lower GFP expression. Still, inducing the *S. tuberosum* pC-HSL receiver with 100 μM pC-HSL resulted in the 10-fold upregulation of *gfp* transcripts. Inducing the *S. tuberosum* pC-HSL receiver with the bacterial sender (*P. putida* pTT337) resulted in the 6-fold upregulation of the *gfp* transcripts (Supplementary Fig. 26).

## Modular sensing and signal processing by engineered bacteria

Once the pC-HSL receiver is put into a plant, the same plant can be made to sense different signals by changing the sensor contained by the bacterium. Multiple sensors can be integrated by genetic circuits performing logic operations that, in turn, control the pC-HSL sender, again relaying this signal to this same plant.

To demonstrate this modularity, strains of *P. putida* were built containing sensors that respond to different chemicals. For proof-of-principle, two sensors were built for small molecules that are often used as inducers: isopropyl β-D-1-thiogalactopyranoside (IPTG, *lacI*[AM]) or anhydrotetracycline (aTc, *tetR*)[93]. The sensors controlling YFP expression were moved from *E. coli* to *P. putida* without making genetic modifications. The response functions of the sensors yielded dynamic ranges of 140-fold (IPTG) and 740-fold (aTc) (Supplementary Fig. 27). The small molecule sensors built for *P. putida* were connected to the pC-HSL sender (Fig. 4a). The bacteria were first grown without inducer and then added to the media containing the plant. They were then induced with 2 mM IPTG or 1 μM aTc, as appropriate. After 24 h, the induction of the pC-HSL receiver in plant roots was quantified (Fig. 4a, Supplementary Fig. 28). The ON and OFF states of the receiver induced by the bacteria were equivalent to the dynamic range observed by the induction with exogenous pC-HSL: 25-fold (IPTG) and 47-fold (aTc).

Next, we demonstrated that *P. putida* and *K. pneumoniae* could act as bacterial sentinels for detecting arsenic. Arsenic is a prevalent and toxic heavy metal that is a global polluter of farmland[101,102]. We constructed an arsenic sensor by expressing the *E. coli* ArsR repressor[103] under control of a strong constitutive promoter ($P_{LacIQ}$)[93] and synthetic RBS[104] to ensure high levels of ArsR. An arsenic responsive promoter was constructed by overlapping the ArsR binding sequence[103] with a strong constitutive promoter (BBa_J23100). The sensor was tested on a pBBR1-ori plasmid in both *P. putida* and *K. pneumoniae*. In *P. putida*, the sensor yielded a 52-fold response and a minimal detection limit of

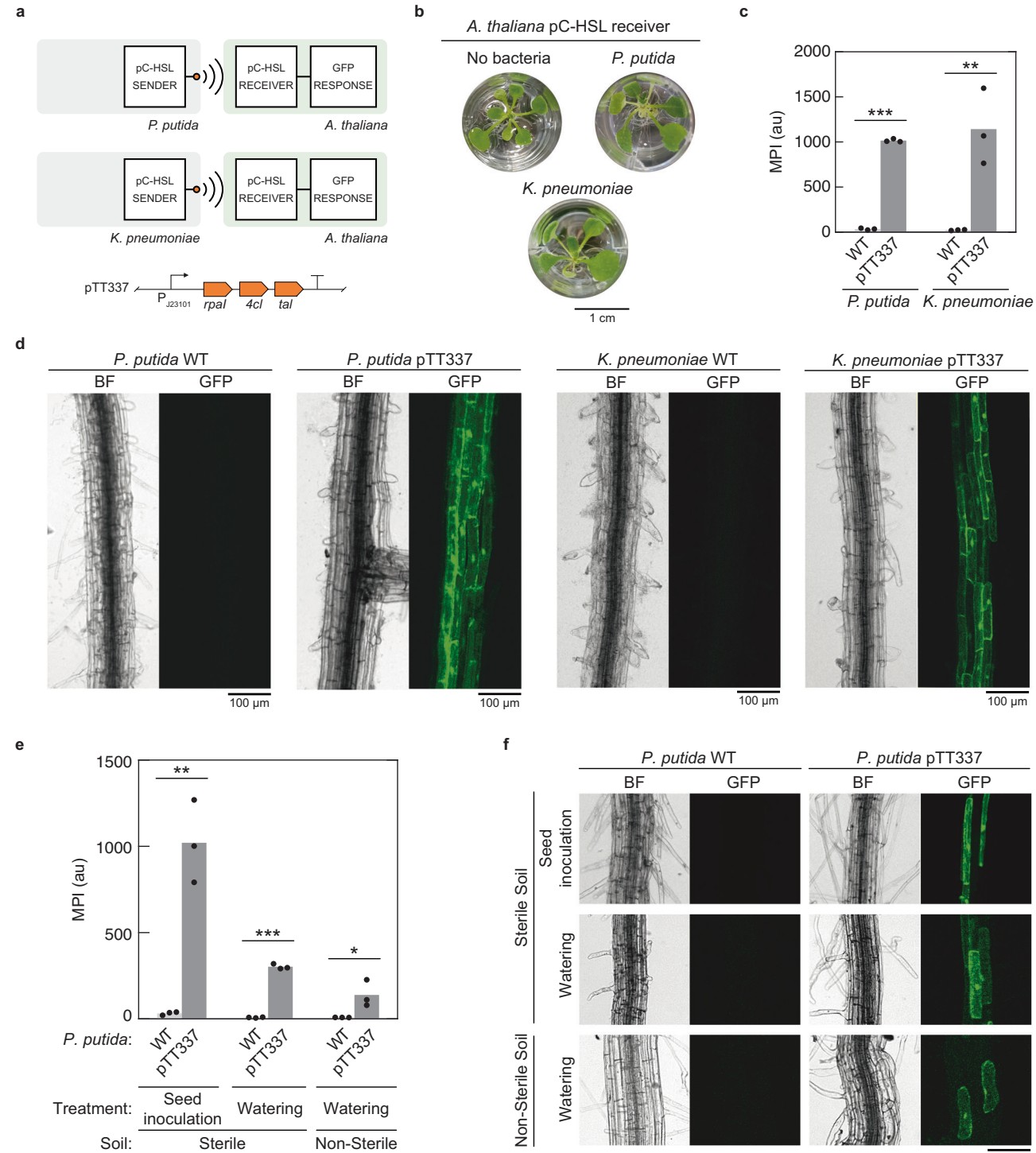

**Fig. 3 | Bacteria-to-plant communication to *A. thaliana* in hydroponics and in soil. a** The constitutive production of pC-HSL by *P. putida* and *K. pneumoniae* was first used to induce the receiver in plants. **b** Phenotypic comparison of *A. thaliana* 315_14_5_1 (Supplementary Table 3) grown in hydroponics with and without wild-type *P. putida* or wild-type *K. pneumoniae* (Methods). **c** The induction of the pC-HSL receiver in *A. thaliana* (*A. thaliana* 315_14_5_1) by *P. putida* or by *K. pneumoniae* constitutively producing pC-HSL (pTT337, Supplementary Fig. 35) in hydroponics (Methods) is shown. The data were extracted from the images in Supplementary Fig. 18. The points were obtained for *n* = 3 plants on different days and the bars represent the means of these points. **d** Induction of the *A. thaliana* pC-HSL receiver in plant roots in hydroponics (Methods). The induction by wild-type *P. putida* (left) and *K. pneumoniae* (right) was compared to when *P. putida* and *K. pneumoniae* constitutively produce pC-HSL (pTT337). Images are representative of experiments performed on three different days with different plants (*A. thaliana* 315_14_5_1). **e** Induction of the *A. thaliana* pC-HSL receiver by *P. putida* constitutively producing pC-HSL (pTT337) in sterile and non-sterile soil (Methods). *P. putida* was introduced either by seed inoculation or through watering (Methods). The data were extracted from the images in Supplementary Fig. 23. The points were obtained for *n* = 3 plants (*A. thaliana* 315_14_5_1) on different days and the bars represent the means of these points. **f** Microscopy images of the induction of the *A. thaliana* pC-HSL receiver by *P. putida* from panel e. Statistical significance was determined using two-tailed Student's *t* test (***$P < 0.001$; **$P < 0.01$; *$P < 0.05$; ns, not significant $P > 0.05$). Source data are provided as a Source Data file.

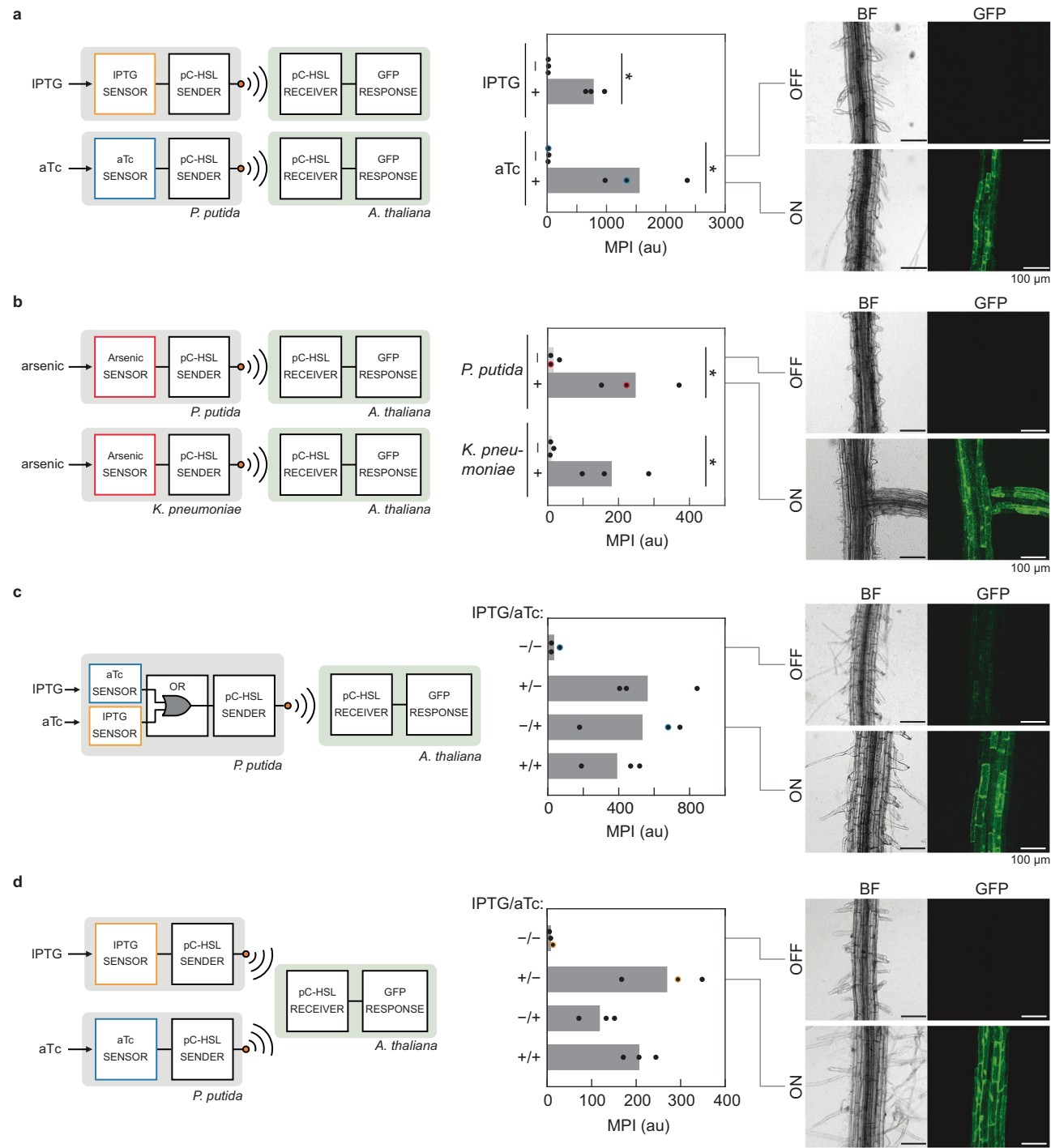

**Fig. 4 | Bacterial sensing and communication to *A. thaliana* and potato. a.** *A. thaliana* responds to *P. putida* engineered to relay pC-HSL upon sensing inducers IPTG or aTc (pTT409 and pTT410, Supplementary Fig. 35) in hydroponics (Methods). The data were extracted from the images in Supplementary Fig. 28. The points were obtained for *n* = 3 plants (*A. thaliana* 315_14_5_1) on different days and the bars represent the means. Microscopy images match the blue-circled replicate. **b.** *P. putida* and *K. pneumoniae* were engineered to detect arsenic (pTT417, Supplementary Fig. 37) and communicate the output to the *A. thaliana* pC-HSL receiver in hydroponics (Methods). The data were extracted from the images in Supplementary Fig. 29. The points were obtained for *n* = 3 plants (*A. thaliana* 315_14_5_1), on different days and the bars represent the means. Microscopy images match the red-circled replicate. **c** *A. thaliana* 315_14_5_1 was co-cultured with *P. putida* sTT659 (Supplementary Table 4) engineered with an OR gate (pTT434, Supplementary

Fig. 38), producing pC-HSL in response to either aTc or IPTG. The data were extracted from the images in Supplementary Fig. 30. Growth conditions and replicates were the same as in part a. *P* values for each of the induced state compared to the uninduced state are: +IPTG/-aTc: 0.02,-IPTG/+ aTc: 0.05, + IPTG/+ aTc: 0.03. Microscopy images match the blue-circled replicate. **d** *A. thaliana* 315 co-cultured with two strains of *P. putida*, each producing pC-HSL in response to a different signal. Strains, growth conditions, and replicates were the same as in part a. The data were extracted from the images in Supplementary Fig. 31. *P* values for each of the induced state compared to the uninduced state are: +IPTG/-aTc: 0.008,-IPTG/+ aTc: 0.010, + IPTG/+ aTc: 0.0007. Microscopy images match the yellow-circled replicate. Statistical significance was determined using two-tailed Student's *t* test (***$P < 0.001$; **$P < 0.01$; *$P < 0.05$; ns, not significant $P > 0.05$). Source data are provided as a Source Data file.

10 ppb and in *K. pneumoniae*, it was 72-fold and 20 ppb (Supplementary Fig. 29). Note these detection ranges are close to the EPA's limit for drinkable water[105]. The arsenic sensor was connected to the pC-HSL sender in *P. putida* and *K. pneumoniae* and GFP expression was present in the roots of the pC-HSL plant receiver: 15-fold and 16-fold induction with *P. putida* and *K. pneumoniae* respectively when induced with 1000 ppb arsenic (Fig. 4b, Supplementary Fig. 29).

Signal processing can modulate the response of sensors, integrate multiple sensors, or implement memory or dynamics[106–108]. While difficult to build in plants, many circuits have been constructed in bacteria, a process simplified with computer aided design (CAD) software[43,47,109–112]. Sensors that have promoters as outputs are easily connected to transcriptional genetic circuits[113]. Logic gates integrate multiple sensors, one of the simplest of which is an OR gate where the output is ON if either of two inputs is ON. Here, we implemented an OR gate in *P. putida* by placing the IPTG-inducible promoter ($P_{Tac}$) and the aTc-inducible promoter ($P_{Tet}$) in series in front of the pC-HSL sender device. The pC-HSL receiver in the Arabidopsis root turned on in the presence of either aTc or IPTG (Fig. 4c and Supplementary Fig. 30). This result demonstrates that the plant can be made to perform computational functions by placing the circuitry in root-associated bacteria.

Multiple bacterial sentinels could transmit information using the same communication channel. As a demonstration, the microbe-to-plant relay experiments were repeated, but with two strains of bacteria, each containing a different sensor connected to the pC-HSL sender (Fig. 4d). Separately, the bacteria were grown in LB medium overnight and then combined into the wells containing the plants. Combinations of the two inducers were added to the wells (1 μM aTc, 2 mM IPTG). Either inducer was sufficient to obtain the induction of the receiver in the root, noting that the magnitudes of the ON states varied for the different combinations of inducers (Fig. 4d, Supplementary Fig. 31). Therefore, the consortium is performing "fuzzy OR" logic.

## Discussion

This work demonstrates a programmable channel of communication from a bacterium to a plant. The advantage of this approach is that the plant receiver only needs to be built and optimized once. Compared to bacteria, new plant sensors are much harder to build and optimize due to slow design-build-test cycles and few genetic parts. As a result, there are far fewer synthetic sensors available for plants and even simple inducible systems are notoriously unavailable. The few that are used are far worse in their performance and dynamic range than what is available for bacteria[114,115]. In contrast, there are hundreds of sensors built for bacteria any of which could be connected to the sender so long as its promoter output has an appropriate dynamic range. Such sensors have been built to respond to toxins, pollutants, nutrients, pathogens, agrochemicals, or any other stimuli[18,106,116,117]. In addition, optimization is much easier in bacteria because it is possible to implement complex selections and there are rapid generation times. The plant receiver allows one plant-based sensor to be developed and optimized, which can then be used with all the sensors and circuits available for bacteria. Changing what is being sensed and the signal processing involves changing the bacterium and not building a new system in the plant.

Complex circuitry could be moved to the bacterium, where it is easier to build[93,108]. Circuits can integrate information from multiple sensors or implements a dynamic response to an input signal (e.g., a pulse)[108]. The circuit's output promoter could be connected to the pC-HSL sender to communicate the result to the plant. Similarly, the pC-HSL receiver was used here to control reporter expression, but it could be connected to metabolic pathways or transcription factors to control morphology[118,119]. Sense-and-respond systems could be distributed, where a bacterial sentinel receives information (e.g., toxin),

transmits it to the plant, which then turns on a response (e.g., detoxification pathway).

Using a common communication signal addresses a second problem in that different molecules will have different uptake and transport properties in the plant. The principal constraint on selecting pC-HSL as a communication signal was that the molecule had to both be producible by a bacterium and able to be sensed by a plant. Transport was also an important consideration, as the molecule must be exported through the prokaryotic membrane and taken up by the root, ultimately entering the nucleus. Prior to selecting pC-HSL, we tested other candidates (data not shown). Lipo-chitooligosaccharides (nod factors) were considered, but we had difficulty both in producing high titers in bacteria and creating a sensor that could be moved to non-legumes. There is an indigo sensor for plants[120] and we could make it at high titer in bacteria using published pathways[121], but we could not export it from the bacterium without lysis. Another mode of inter-kingdom signaling is the production of plant hormones, such as auxins, by growth-promoting bacteria[122]. However, these have extensive effects on plant growth and gene expression and are ubiquitous across plant species. Using the same plant promoter scaffold described in this work, we also attempted to make sensors for DAPG (PhlF), OC6-HSL (LuxR) and OC8-HSL (TraR), but they failed.

More generally, rare HSLs may be the ideal molecules for inter-kingdom signaling. Homoserine lactones can transport across the walls of many cell types, can be produced at sufficient titer with a few enzymes, are non-toxic and bind to well-defined regulators. Note that while the HSL signal from the bacteria was detectable in non-sterile soil, it was weaker compared to sterile soil conditions, and factors such as the water source and composition, microbial content, soil treatment, and specifics to the system we use introduce variables that could attenuate signal strength and affect plant phenotype. These considerations are likely to lead to less predictability under field conditions. This reduced activity could be due to the microbes degrading or sequestering the signaling molecule. Thus, acyl-HSLs may be too common to enable specificity between an engineered plant and microbe and have too systematic of impact on plant gene expression. However, there are many other HSL structures from which to choose. This includes unusual branched-chain groups, other aroyl- groups (e.g., cinnamoyl-HSL) and structurally unrelated mimicking compounds[9,123,124]. Non-natural HSLs have been built with synthetic organic chemistry, including sulfonyl, aroyl and alkanoyl-HSLs[81,125]. All natural homoserine lactones are L-isomers and D-isomers are not biologically active in plants[126,127]. Synthetic pathways to the D-isomers could reduce crosstalk with plant signaling, but this would require engineering new regulators. Finally, retrosynthetic design software[128] could create completely new chemistries that are not degraded by native bacteria.

Microbes beneficial to crops have long been used in agriculture. Typically, one species has been used or few have been combined into an artificial consortium, relying on the capabilities of unmodified species. Building such consortia has been ad hoc, and it has been noted that the positive effects of multiple species often do not combine additively[62,129]. Advances in genetic engineering make the design of multi-species consortia and the rational distribution of functions across the consortia possible. This work lays the foundation for using bacteria grown in proximity to a plant to perform tasks such as monitoring soil nutrient content, sensing pathogens, or detecting environmental contaminants.

In addition to agricultural applications, this approach could be applied to build plant sentinels. For example, it has been proposed to use engineered plants to detect landmines by sensing TNT[130]. Moving the sensing to root-associated bacteria allows the same plant to be used to detect different signals simply by swapping the engineered bacterium with which it is partnered that contains the new sensor. Further, when combined with the plant-to-microbe signal developed

previously[38], this completes two-way communication to coordinate interkingdom functions, such as the establishment of synthetic symbiosis[131,132]. By viewing the plant-microbe community holistically, we can select the organism best suited for a particular task and use orthogonal channels of communication to stabilize and coordinate the population. Desirable functions could be distributed amongst members of the consortium, where individual species are assigned duties such as detoxification, pathogen defense and nutrient scavenging.

## Methods

### Strains, media and chemicals

All bacterial strains are listed in Supplementary Table 4. *Escherichia coli* NEB® 10-beta (New England BioLabs, C3019I) was used to clone all plasmids. *Pseudomonas putida* KT2440 (ATCC 47054) and *Klebsiella pneumoniae* 342 (ATCC BAA-2552) were used for co-culture experiments. *Agrobacterium tumefaciens* GV3101 (Gold Bio, GV3101 Electrocompetent) was used for the floral dip method. Bacterial cells were routinely grown in LB Miller broth (Difco, 244620) at 37 °C for *E. coli* and 30 °C for *P. putida* and *A. tumefaciens*. Plates were made using LB Miller broth with 1.5% Bacto Agar (Difco, 214010). Antibiotics were used to maintain plasmids during routine growth: kanamycin (GoldBio, K-120-10)-35 μg/mL for *E. coli*, 50 μg/mL for *P. putida* and *A. tumefaciens;* tetracycline (GoldBio, T-101-25)-10 μg/mL for *E. coli*, 25 μg/mL for *P. putida*; gentamycin (Enzo Life-sciences, 380-003-G001)-15 μg/mL for *E. coli*, 50 μg/mL for *P. putida*; chloramphenicol (Alfa Aesar, B20841)-25 μg/mL for *P. putida;* rifampicin (Santa Cruz Biotechnology, SC-200910)-50 μg/mL for *A. tumefaciens*. Stocks of 10 mM HSLs were solubilized in DMF and stored at -20 °C: N-3-oxohexanoyl-L-homoserine lactone (OC6-HSL; Sigma, K3007); 3-hydroxytetradecanoyl-homoserine lactone (OHC14-HSL; Sigma, 51481); N-3-oxododecanoyl-L-homoserine lactone (OC12-HSL; Sigma, O9139); *p*-coumaroyl-homoserine lactone (pC-HSL; Sigma, 07077). Bacterial cells were induced using the following chemical stocks: IPTG (isopropyl-ß-D-1-thiogalactopyranoside; Gold Biotechnology, I2481) in water; aTc (anhydrotetracycline; Sigma, 37919) in 50% (v/v) ethanol; arsenic (sodium meta-arsenite; Sigma, S7400) in water. For flow cytometry, cells were diluted in phosphate buffer saline (PBS; Sigma, 6505-4 L). MSVI vitamin solution (2 mg/mL glycine -Thermo Scientific, J16407-36; 0.5 mg/mL nicotinic acid - Sigma, N0761-100G; 0.5 mg/mL pyridoxine HCl - Sigma, P2680-25G; 0.4 mg/mL thiamine HCl - Sigma, T1270-25G), JHMS vitamin solution (0.4 mg/mL folic acid - Phytotech, F430; 0.05 mg/mL biotin - Phytotech, B140) and 3R vitamins solution (1 mg/mL thiamin HCl - Sigma, T1270-25G; 0.5 mg/mL nicotinic acid - Sigma, N0761-100G; 0.5 mg/mL pyridoxine HCl - Sigma, P2680-25G) were used for the preparation of CIM and 3C5ZR media.

### *A. thaliana* growth and transformation

All plant lines are listed in Supplementary Table 3. *A. thaliana* Col-0 seeds (NACS, CS70000) were acquired from the Arabidopsis Biological Resource Center. When working with a small number of seeds, they were surface sterilized with 70% ethanol for 1–2 min, followed by 10% bleach for 10 min, and then rinsed 5 times with water. Larger numbers of seeds were sterilized using the chlorine gas method[133]. Sterilized seeds were sown on half strength Murashige and Skoog (Sigma, M5519) media with 1% sucrose (Fisher Scientific, S5-3) and adjusted to pH 5.7 with KOH (MS media). Plates were made with the addition of 0.8% agar (Sigma, A7921). Plates were sealed with Micropore™ tape (3M™, 1530-0) to allow for gas exchange. Seeds were stratified at 4 °C in the dark for 3 days before moving to a growth chamber (Percival Scientific, CU-36L5) where they were grown at 27 °C in 16/8 h light/dark cycles with a light intensity of 40 μmol/m²/s. For growth to seed, plants were also grown in soil in a greenhouse with 16/8 h light/dark cycles at 21 °C. Transgenic *A. thaliana* lines were generated by *Agrobacterium tumefaciens*-mediated floral dip. Briefly, *A. tumefaciens* strains containing plasmids of interest were cultured in 2 mL LB media containing appropriate antibiotics at 30 °C and 250 r.p.m. for 2 days (Brunswick Scientific, Innova 44). This culture was used to inoculate 500 mL LB media with appropriate antibiotics and cultured for an additional 24 h. Cultures were then pelleted by centrifugation at 4000 × g for 10 min at 4 °C. Pellets were resuspended by pipetting with a serological pipette in 5% (w/v) sucrose solution plus 0.02% (v/v) Silwet L-77 (Phytotech Labs, S7777). Arabidopsis inflorescences were submerged in the bacterial resuspension for 1 min with gentle agitation, removed and drip dried, and covered gently in plastic wrap before being transferred to the dark overnight. The next day transformed plants were returned to the greenhouse until they produced seeds. T1 seeds were sown on moistened soil and covered with clear plastic lid until cotyledons were visible at which point the lid was removed and the seedlings were sprayed with 50 μM of PPT (Phosphinothricin; GoldBio, P-165-1) twice per week for three weeks until only resistant lines remained. Transgenic T1 plants were grown to seed. T2 seeds were sown on agar plates containing 50 μM PPT. Resistant seedlings were transferred to soil at three weeks and grown to seed. T3 seeds were sown on agar plates containing 50 μM PPT and stable, homozygous lines were validated by segregation.

### Arabidopsis phenotypic analysis

The pC-HSL receiver and wild-type Arabidopsis lines were transferred from tissue culture to 6 × 6 × 9 cm pots with Pro-Mix BK25 potting mix (Griffin Greenhouse Supplies, 94-1110). Plant height, number of rosette leaves, number of primary shoots, fresh weight was collected at time of flowering. Leaves and stems were dried in an oven at 60 °C for 1 week and weighed.

### Arabidopsis root phenotypic analysis

The pC-HSL receiver and wild-type Arabidopsis seeds were germinated and grown on MS media with 1% sucrose. Seeds were stratified for 3 days at 4 °C, then petri plates were placed vertically in a growth chamber with light intensity at 90 μmol/m²/s, 16/8 h light/dark cycle, and a temperature of 24 °C. Eleven days after sowing seeds, plates were imaged on a ChemiDoc MP Imaging System (Bio-Rad) using a white tray. Images were taken with a 590/110 nm filter at 0.5 s exposure. Root length was measured using ImageJ segmented line tool. Lateral roots were counted from plates, and lateral root density is calculated by dividing lateral roots by total primary root length.

### Chemical HSL induction of *A. thaliana* receivers in the hydroponic system

See Supplementary Fig. 1a. Seeds were surface sterilized with 70% ethanol for 1–2 min, followed by 10% bleach for 10 min, and then rinsed 5 times with water. They were then suspended in 300 μL of 0.1% agar (Sigma, A7921) and sown at 1 cm intervals onto square Petri dishes (Fisherbrand, FB0875711A) of half strength Murashige and Skoog (Sigma, M5519) medium with 1% sucrose, 0.8% agar (Sigma, A7921) and adjusted to pH 5.7 with KOH (MS medium). The plates were placed in the dark at 4 °C for a 3-day striation period before being moved to a growth chamber (Percival Scientific, CU-36L5) and grown for 7–12 days. In a tissue hood, individual wells of 24-well plates (Falcon, 353047) were filled with 1 mL MS medium. HSLs were added to appropriate wells. Finally, plants were carefully lifted from the agar plates with forceps and moved to individual wells such that the roots were entirely submerged. Each plate was covered with a lid, sealed with Micropore™ tape, and returned to the growth chamber. After 24 h, plates were taken from the growth chamber for imaging.

### Confocal microscopy

Microscopy experiments were performed using a Nikon A1R Ultra-Fast Spectral Scanning Confocal Microscope and Andor iXON EMCCD camera or the Leica SP8 Laser Scanning Confocal Microscope. A

4X/0.20 Plan Apo or a 10X/0.30 Plan Fluor air objective was used with the Nikon microscope and a Fluotar Visir 25X/0.95 water objective or a HC PL APO CS2 10x/0.40 air objective was used with the Leica SP8 microscope. For the Nikon microscope, the fluorescence signal was visualized with an excitation wavelength of 488 nm and emission wavelength of 525 nm. To enable comparisons between different days and plant lines, we used the same laser intensities and microscope settings for all experiments performed with the Nikon microscope: the 488 nm laser was used at 10% power, HV 100, and 0 offset. For the Leica SP8 microscope, GFP was visualized with the 488 nm laser at 2% power, 100% gain, emission was detected with the HyD 1 detector between 493 and 529 nm, with 2–4 times line average. Propidium iodide (PI) was visualized with the 534 nm laser at 5% power, 44.3% gain, emission was detected with the HyD 3 detector between 540 and 742 nm, with 2–4 times line average. Upon loading each sample, the entire root system of each plant was inspected, and images were captured only of the brightest portion. For spatiotemporal imaging with the Leica SP8, Arabidopsis roots were stained with 10 µg/ml propidium iodide (Invitrogen, P1304MP) in water. All images were analyzed using the FIJI package of ImageJ. Bright field and fluorescent images were aligned using the Landmark Correspondences plugin. For display purposes, the LUT of the fluorescent channels were inverted in composite images. To make the figures, the minimum and maximum brightness of the LUT were adjusted using the ImageJ default Brightness tool. Note that only the unaltered images were used for MPI quantification. A custom macro was used for fluorescence quantification (Supplementary Fig. 2). A Gaussian filter ($\sigma = 2$) was used for noise reduction followed by auto-thresholding using Otsu's method to separate root tissue from background. The binary mask was then applied to the original image, selecting only the root sections. The mean pixel intensity of just the roots ($MPI_{roots}$) was calculated using the Measure tool in ImageJ. Finally, the mean pixel intensity of the background ($MPI_{background}$) was calculated and subtracted from the $MPI_{roots}$ to get the final MPI used in the paper.

### Quantitative real time PCR analysis for Arabidopsis
Total RNAs were extracted from the seedling with the RNeasy Plant Mini Kit (Qiagen, 74904). cDNAs were generated by the High-Capacity cDNA Reverse Transcription Kit (ThermoFisher Scientific, 4368814). Transcript levels were amplified with the primers listed in Supplementary Table 1 and the reactions were set up by SYBR™ Select Master Mix (Thermo Fisher Scientific, A46109). ACTIN 2 was used as an internal control to normalize expression levels. Quantitative real-time PCRs were performed by the LightCycler 480 (Roche). The sequences of the target gene (primers) are: sfGFP (GAGGGTGAAGGTGACGCAACTAATG, GGACTTGAAGAAGTCATGCT GCTTC) and actin2 housekeeping gene (GTCGTACAACCGG TATTGTGCTG, CCTCTCTCTGTAAGGATCTTCATGAG).

### Response functions
Data from multiple plants or multiple bacterial cultures were used to fit a Hill function:

$$y = y_{min} + (y_{max} - y_{min}) \left( \frac{x}{\kappa + x} \right)^n \tag{1}$$

where $y$ is the output (au), $x$ is the concentration of inducer, $\kappa$ is the threshold concentration, and $n$ is the cooperativity (Supplementary Table 1).

### Soil preparation
Soil was prepared by mixing three scoops (Grainger, REMCO 82 oz hand scoop, 3UE74) of soil (Lambert Peat Moss Inc, LM-2 Germination Mix, 664980-2325), 1 scoop (Grainger, REMCO 82 oz hand scoop, 3UE74) of vermiculite (Griffin, Whittemore D3 Fine, 65-3120) and one

small scoop (30 mL) of Osmocote (The Scotts Miracle-Gro Company, Osmocote 14-14-14, 277960). Soil was sterilized by autoclaving.

### Chemical HSL induction of the *A. thaliana* pC-HSL receiver in soil
See Supplementary Fig. 1c. Seeds were surface sterilized with 70% ethanol for 1–2 min, followed by 10% bleach for 10 min, and then rinsed 5 times with water. They were then suspended in 300 µL of 0.1% agar (Sigma, A7921) and sown at 1 cm intervals onto square Petri dishes (Fisherbrand FB0875711A) of half-strength Murashige and Skoog (Sigma, M5519) medium with 1% sucrose, 0.8% agar (Sigma, A7921) and adjusted to pH 5.7 with KOH (MS media). The plates were placed in the dark at 4 °C for a 3-day striation period before being moved to a growth chamber under the conditions described above and grown for 5 days until the emergence of the first leaf. The soil mixture, prepared as described above, was thoroughly wet with sterile tap water, mixed with a spatula (Cole-Parmer, 17211), and used to fill 10 mL glass beakers (Pyrex, 1000-10). The soil mixture was lightly compressed to provide a firm bed for the seeds. Next, individual seedlings were transplanted into the beakers with forceps. The beakers were placed inside deep Petri dishes (Sigma, P5606-400EA), sealed with Micropore™ tape, and incubated for 10 days in the growth chamber (Percival Scientific, CU-36L5). After 5 days in the growth chamber, the seedlings were watered by adding 1 mL of autoclaved MilliQ water to each beaker by pipetting and placed back into the growth chamber. After 10 days in the growth chamber, each plant was induced with autoclaved MilliQ water or pC-HSL by watering the plant by pipetting at the plant-soil interface with 1 mL of autoclaved MilliQ water and 1 mL of 100 µM pC-HSL respectively. We estimate the effective concentration of pC-HSL in the soil to be 260 nM (average molarity = (total moles of pC-HSL added)/(total water in soil) = ($10^{-6}$ M x $10^{-3}$ L)/(((4.5 g wet soil-1.74 g dry soil)/1000 g/L) + 1 mL) = 260 nM). The plants were placed back into the growth chamber and incubated for 24 h. After 24 h, the plants roots were cleaned with tap water to remove the soil for imaging.

### Genome analysis
To search for HSL producing genes in the *P. putida* KT2440 (Taxonomy ID: 160488) and *K. pneumoniae* 342 (Taxonomy ID: 507522) genomes were performed using the protein basic local alignment search tool (BLASTp; blast.ncvi.nlm.nih.gov) using the blastp (protein-protein BLAST) program and default parameters. The query genes were *luxI* (*Vibrio fischeri*, NCBI-Protein ID: AAW87994), *cinI* (*Rhizobium leguminosarum*, NCBI-Protein ID: WP_018242930), *lasI* (*Pseudomonas aeruginosa*, NCBI-Protein ID: QPV56976), *traI* (*Agrobacterium tumefaciens*, NCBI-Protein ID: WP_010974838), and *rpaI* (*Rhodopseudomonas palustris*, NCBI-Protein ID: WBU30219) which all resulted in no significant similarity found (no sequences were returned).

### Potato growth and transformation
*Solanum tuberosum* (potato) var. 'Desirée' was grown in Magenta GA7 vessels (Thomas Scientific, 1190X31) with solid MS Reg medium (4.33 g/L MS basal salt mixture - Phytotech, M524; 25 g/L sucrose - Fisher Science Education, S25590B; 100 mg/L myo-inositol - Sigma, I7508-500G; 170 mg/L sodium phosphate monobasic monohydrate - Sigma, S9638-500G; 440 mg/L calcium chloride dihydrate - Sigma, C7902-500G; 0.9 mg/L thiamine-HCl - Sigma, T1270-25G; 2 mg/L glycine - Thermo Scientific, J16407-36; 0.5 mg/L nicotinic acid - Sigma, N0761-100G; 0.5 mg/L pyridoxine-HCl - Sigma, P2680-25G; 1 × MS vitamins - Phytotech, M553; 3 g/L Phytagel™ - Sigma, P8169-250G; pH 5.7 adjusted with KOH - Sigma, P2680-25G) under fluorescent lights (Sylvania, F34CW/SS/ECO-light intensity 70 m$^{-2}$ s$^{-1}$) at ambient temperature[134]. The pC-HSL receiver plasmid was modified to replace the phosphinothricin resistance cassette with one for hygromycin resistance for selection in potato. The pTT315-Hyg plasmid (Supplementary Fig. 34) was then transformed into *A. tumefaciens* LBA4404

with the freeze-thaw method[135]. For potato nuclear transformation, *A. tumefaciens* cultures were grown to OD$_{600}$ = 0.6, centrifuged, and resuspended in liquid CIM medium (4.3 g/L MS salt - Polytech, M524; 1 mL/L MSVI vitamins; 1 mL/L JHMS vitamins; 0.1 g/L inositol - Sigma, I7508-500G; 30 g/L sucrose - Fisher Science Education, S25590B; 1 mg/L 6-benzylamino purine (BAP) - Sigma, B3408; 2 mg/L 1-naphthalene-acetic acid (NAA) - Phytotech, N605; 10 g/L agar - BD, 214010; pH 5.7 adjusted with KOH - Sigma, P2680)[136]. One-month old potato 1 cm internodes were placed on solid CIM media in petri plates with 20 mL of resuspended *A. tumefaciens*[136]. After 20 min, internodes were transferred to new solid CIM medium and placed in the dark[136]. After 48 h, internodes were transferred to 3C5ZR medium (4.3 g/L MS salt - Phytotech, M524; 1 mL/L 3R vitamins solution; 0.1 g/L inositol - Sigma, I7508-500G; 30 g/L sucrose - Fisher Science Education, S25590B; 10 g/L agar - BD, 214010; 0.5 mg/L 3-indoleacetic acid (IAA) - Phytotech, I364; 3 mg/L trans-zeatin-riboside - Phytotech, Z875; 500 mg/L timentin - Phytotech, T869 added after sterilization; pH 5.7 adjusted with KOH - Sigma, P2680-25G) until shoots formed, which were transferred to MS Reg media with 20 mg/L hygromycin (Phytotech, H397) and 200 mg/mL timentin (Phytotech, T869) in Magenta vessels. Five lines were regenerated and confirmed for transgene integration by PCR.

### Quantitative real time PCR analysis for potato
Each line was propagated into new media until root formation, and then induced with either pC-HSL or *P. putida* as described for Arabidopsis. After 24 h, RNA was isolated from roots. cDNA was synthesized from 500 ng of RNA for each plant using ZymoScript RT PreMix Kit (Zymo Research, R3012). qRT-PCR was conducted with PowerUp SYBR Green Master Mix (Thermo Scientific, A25741) on a QuantStudio 3 (Thermo Fisher Scientific) using 1 μL of cDNA. The sequences of the target (primers) are: sfGFP (CTCCAATCGGTGATGGTCCT, GCAGAAC CATATGATCGCGT), VP16 domain (TGGACATGTTGGGGGACGG, CTC GAAGTCGGCCATATCCAG), and the Ef1α housekeeping gene (GATGG TCAGACACGTGAACA, CCTTGGAGTACTTGGGGGTG).

### Potato phenotypic analysis
The receiver and wild-type potato lines were transferred from tissue culture to 6 × 6 × 9 cm pots with Pro-Mix BK25 potting mix (Griffin Greenhouse Supplies, 94-1110). After 4 weeks, plants were transferred to 11.4 L pots and grown until bolting in a greenhouse. Plant height, fresh weight of leaves and stems, and chlorophyll content were collected at time of bolting. Chlorophyll content was measured with a CCM-200 plus Chlorophyll Content Meter (Opti-Sciences). Leaves and stems were dried in an oven at 60 °C for 1 week and weighed.

### pC-HSL sensor characterization in *E. coli*
Single colonies of *E. coli* MG1655 sTT658 (Supplementary Table 4) were inoculated into 1 mL LB medium with antibiotics in 2 mL 96 deep-well plates (Thermo, AB-0788), sealed with AeraSeal film (Excel Scientific, BS-25), and grown at 30 °C at 900 rpm. (INFORS HT, Multitron Pro). A 0.5 μL aliquot of overnight culture was diluted into 150 μL LB medium with antibiotics and pC-HSL inducer in 96-well V-bottom plates (Thermo, 249952) and grown at 30 °C at 1000 rpm for 3 h in an EMLI shaker (ELMI, DTS-4). A 3 μL aliquot of the culture was diluted into 200 μL PBS in a round-bottom 96-well plate (Corning, 3797) and analyzed using cytometry.

### Flow cytometry
A BD LSR Fortessa flow cytometer with High Throughput Sampler (HTS) attachment (BD Biosciences). At least 10,000 events were captured for each sample and gated by forward and side scatter. Measurements were made using a FITC channel voltage of 450 V, PE-TexasRed channel voltage of 600 V, an FSC voltage of 640 V, and SSC voltage of 289 V. FlowJo and Cytoflow (cytoflow.github.io) were used for analysis and gating. The median fluorescence values were reported.

### Measurement of pC-HSL production from *P. putida* and *K. pneumoniae*
*P. putida* pTT337 or *K. pneumoniae* pTT337 was streaked onto LB agar plates with antibiotics and grown overnight at 30 °C or 37 °C for *P. putida* and *K. pneumoniae* respectively. Individual colonies were inoculated into 1 mL of LB medium with appropriate antibiotics in 2 mL 96 deep-well plates grown overnight at 30 °C at 900 rpm (INFORS HT, Multitron Pro). The next morning, the OD$_{600}$ was measured in a 1 mL cuvette (VWR, 97000-586) in a spectrometer (Agilent Technologies, Cary 60 UV-Vis). The culture was diluted to OD$_{600}$ = 0.01 into 1 mL of 100% MS medium supplemented with 100 μM of *p*-coumarate with or without *A. thaliana* pC-HSL receiver, and grown at 27 °C without shaking for 24 h in the growth chamber (Percival Scientific, CU-36L5). The culture was transferred to 2 mL microfuge tubes and centrifuged at 9600 × g for 5 min. The supernatant was moved to a 96-well filter plate (Whatman, Unifilter 800) and centrifuged at 2200 × g for 5 min. Dilutions of the filtered supernatant were used to induce *E. coli* MG1655 sTT658 (Supplementary Table 4) in 150 μL LB medium. The *E. coli* cultures were grown at 30 °C at 1000 r.p.m. for 3 h in an EMLI shaker (ELMI, DTS-4). The cells were diluted into PBS in a round-bottom 96-well plate (Corning, 3797) and analyzed using cytometry. Using the response function of the *E. coli* pC-HSL sensor strain (*E. coli* MG1655 sTT658, Supplementary Fig. 19) as a calibration curve, the YFP fluorescence from *E. coli* MG1655 sTT658 grown in the *P. putida* or *K. pneumoniae* supernatants was compared to the YFP fluorescence of the calibration curve (Supplementary Fig. 19). The pC-HSL concentrations of the diluted supernatants was determined by regression.

### Bacterial induction of *A. thaliana* HSL receiver in the hydroponic system
See Supplementary Fig. 1a. Seeds were surface sterilized with 70% ethanol for 1–2 min, followed by 10% bleach for 10 min, and then rinsed five times with water. They were then suspended in 300 μL of 0.1% agar (Sigma, A7921) and sown at 1 cm intervals onto square Petri dishes (Fisherbrand, FB0875711A) of half strength Murashige and Skoog (Sigma, M5519) media with 1% sucrose, 0.8% agar (Sigma, A7921) and adjusted to pH 5.7 with KOH (MS media). The plates were placed in the dark at 4 °C for a 3-day striation period before being moved to a growth chamber under the conditions described above and grown for 7–12 days. Two days before plant inoculation, *P. putida or K. pneumoniae* strains were streaked from glycerol stocks onto LB agar plates with appropriate antibiotics and grown at 30 °C or 37 °C overnight for *P. putida* and *K. pneumoniae* respectively. The next day, individual colonies were selected and inoculated into 1 mL LB medium with appropriate antibiotics in 96 deep-well plates. These plates were grown overnight at 30 °C or 37 °C at 900 r.p.m. (INFORS HT Multitron Pro). The day of plant inoculation, the cultures were spun down for 3 min at 6100 × g, the supernatant was discarded, and the pellet was resuspended in 2 mL of MS medium. This rinsing step was repeated twice to ensure removal of pC-HSL produced overnight. The OD$_{600}$ was measured in a 1 mL cuvette (VWR, 97000-586) in a spectrometer (Agilent Technologies, Cary 60 UV-Vis). In a tissue hood, individual wells of 24-well plates (Falcon, 353047) were filled with 1000 μL of MS media. *P. putida* or *K. pneumoniae* was inoculated in the wells to a starting OD$_{600}$ of 0.01. For *P. putida* containing inducible control of HSL production from individual sensors, appropriate chemical inducers were added at the following concentrations: 2 mM IPTG; 1 μM aTc; 1000 ppb. For *K. pneumoniae* containing the arsenic sensor, a starting OD$_{600}$ of 0.4 was used and arsenic was added at 1000 ppb. *p*-coumarate (Sigma, C9008) (100 μM) was also added. Finally, plants were lifted from the agar plates with forceps and moved to individual wells such that the roots were entirely submerged. Each plate was covered with a lid, sealed with Micropore™ tape, and returned to the growth chamber. After 24 h, plants were removed from each well and placed on a microscope slide (VWR, Micro Slides

48300-026) under a 22x40mm No 1 cover slip (VWR, Cover glass 48393-048) for imaging.

## MS agar induction of the *A. thaliana* pC-HSL receiver

See Supplementary Fig. 1b. Seeds were surface sterilized with 70% ethanol for 1–2 min, followed by 10% bleach for 10 min, and then rinsed 5 times with water. They were then suspended in 300 μL of 0.1% agar (Sigma, A7921) and sown at 1 cm intervals onto square Petri dishes (Fisherbrand, FB0875711A) of half strength Murashige and Skoog (Sigma, M5519) media with 1% sucrose, 0.8% agar (Sigma, A7921) and adjusted to pH 5.7 with KOH (MS media). The plates were placed in the dark at 4 °C for a 3-day striation period before being moved to a growth chamber (Percival Scientific, CU-36L5) and grown for 6 days. MS agar plates sectioned into quadrants (VWR, 25384-348) were prepared, such that the top quadrants would be filled with 1% MS agar without inducer and the bottom quadrants would be filled with MS agar supplemented with either 0 or 100 μM pC-HSL for chemical induction of the *A. thaliana* pC-HSL receiver. For induction with *P. putida*, the lower quadrants were filled with 1% MS agar supplemented with 100 μM *p*-coumarate and $OD_{600} = 0.01$ of *P. putida* WT or *P. putida* pTT337. Before measuring the $OD_{600}$, the bacterial cultures were spun down for 3 min at $6100 \times g$, the supernatant was discarded, and the pellet was resuspended in 2 mL MS medium. This rinsing step was repeated twice to ensure removal of pC-HSL produced overnight. In a tissue hood, the 7-day old seedlings were transferred onto these new agar plates, carefully laying the leaves onto the top quadrants such that no leaf would be in direct contact with either pC-HSL or *P. putida* and laying the roots onto the bottom quadrants with or without pC-HSL inducer/*P. putida*. Plates were incubated in the growth chamber for another 24 h into stands to allow the plants to grow vertically. After 24 h, plates were taken from the growth chamber for imaging into the Leica SP8 Laser Scanning Confocal Microscope or from *gfp* transcript quantification using RT-qPCR.

## Bacterial induction of the *A. thaliana* pC-HSL receiver by seed inoculation in soil

See Supplementary Fig. 1d. Seeds were surface sterilized with 70% ethanol for 1–2 min, followed by 10% bleach for 10 min, and then rinsed 5 times with water. They were then suspended in 300 μL of 0.1% agar (Sigma, A7921) and kept into the dark at 4 °C for a 3-day striation period. Sterile and non-sterile soil mixtures, prepared as described above, were thoroughly wet with sterile tap water, mixed with a spatula (Cole-Parmer, 17211), and used to fill 10 mL glass beakers (Pyrex, 1000-10). The soil mixture was lightly compressed to provide a firm bed for the seeds. After the striation period, the seeds were incubated for 1 h with *P. putida* WT or *P. putida* pTT337 grown overnight at 30 °C in LB medium at 900 rpm (INFORS HT, Multitron Pro). The seeds were then transferred to soil (either sterile or non-sterile). The glass beakers were covered with Saran wrap, held into place with elastic bands and 3 puncture holes were made with the tip of sterile forceps to allow for aeration in the beaker. Plants were then incubated for 20 days in the growth chamber (Percival Scientific, CU-36L5) and they were watered with autoclaved MilliQ water supplemented with 100 μM of *p*-coumarate at day 5 and day 15. After 20 days, the plants roots were cleaned with tap water to remove the soil for imaging.

## Bacterial induction of the *A. thaliana* pC-HSL receiver by watering in soil

See Supplementary Fig. 1e. Seeds were surface sterilized with 70% ethanol for 1–2 min, followed by 10% bleach for 10 min, and then rinsed 5 times with water. They were then suspended in 300 μL of 0.1% agar (Sigma, A7921) and kept into the dark at 4 °C for a 3-day striation period. Sterile and non-sterile soil mixtures, prepared as described above, were thoroughly wet with sterile tap water, mixed with a spatula (Cole-Parmer, 17211), and used to fill 10 mL glass

beakers (Pyrex, 1000-10). The soil mixture was lightly compressed to provide a firm bed for the seeds. After the striation period, the seeds were sowed into either sterile or non-sterile soil and incubated for 5 days in the growth chamber. After 4 days, *P. putida* WT or *P. putida* pTT337 were grown overnight in LB medium supplemented with antibiotics at 30 °C in LB medium at 900 rpm (INFORS HT, Multitron Pro). The next day, the cultures were spun down by centrifugation for 3 min at $6100 \times g$ and resuspended in 2 mL of PBS. The wash step was repeated twice. The $OD_{600}$ was measured in a 1 mL cuvette (VWR, 97000-586) in a spectrometer (Agilent Technologies, Cary 60 UV-Vis). The culture was diluted to $OD_{600} = 1$ into PBS for a final volume of 2 mL. The 2 mL of the diluted culture were supplemented with 100 μM of *p*-coumarate was added to the soil. At day 15, this was repeated. After 20 days, the plants roots were cleaned with tap water to remove the soil for imaging.

## Characterization of bacterial genetically-encoded sensors

Overnight cultures from individual colonies were grown in 1 mL LB media with antibiotics in 96 deep-well plates (Thermo, AB-0788). A 3 μL aliquot of overnight culture was diluted into 150 μL LB medium with antibiotics and inducer in 96 well V-bottom plates (Thermo, 249952) and grown at 30 °C at 1000 rpm for 4.5 h in an EMLI shaker (ELMI DTS-4). A 3 μL aliquot of the culture was diluted into 200 μL PBS in a round-bottom 96-well plate (Corning, 3797) and analyzed using cytometry.

## Reporting summary

Further information on research design is available in the Nature Portfolio Reporting Summary linked to this article.

## Data availability

The *P. putida* KT2440 (Taxonomy ID: 160488) and *K. pneumoniae* 342 (Taxonomy ID: 507522) genomes were used to search for homologs of *luxI* (*Vibrio fischeri*, NCBI-Protein ID: AAW87994), *cinI* (*Rhizobium leguminosarum*, NCBI-Protein ID: WP_018242930), *lasI* (*Pseudomonas aeruginosa*, NCBI-Protein ID: QPV56976), *traI* (*Agrobacterium tumefaciens*, NCBI-Protein ID: WP_010974838), and *rpaI* (*Rhodopseudomonas palustris*, NCBI-Protein ID: WBU30219) query genes using the protein basic local alignment search tool BLASTp [https://blast.ncbi.nlm.nih.gov] with the blastp (protein-protein BLAST) program and default parameters. Plant genetic parts are available as part of Supplementary Data 1. Bacterial genetic parts are available as part of Supplementary Data 2. Plasmid maps are available as part of Supplementary Data 3. Movie files from Supplementary Fig. 9 are available as Supplementary Movies 1–4. All microscopy images used for fluorescence quantification are available in the Supplementary Information at a resolution of 600 dpi and the raw TIFF files are available at Zenodo [https://doi.org/10.5281/zenodo.10601326]. Source data are provided with this paper.

## Code availability

The custom Fiji script used for the mean pixel intensity (MPI) calculation is available at Github [https://github.com/VoigtLab/plant-microbe-communication].

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

## Acknowledgements

We thank the Whitehead Institute and Jing-Ke Weng for access to the greenhouse facility and technical support. We also thank the Microscopy Core at the Koch Institute's Robert A. Swanson (1969) Biotechnology Center, the Barbara K. Ostrom (1978) Bioinformatics Facility, Jeffrey Kuhn, Jeff Wyckoff and Duanduan Ma for technical support in acquiring and analyzing confocal microscopy images. This material is based upon work supported by the National Science Foundation Graduate Research Fellowship grant #1745302 (T.T.), the Defense Advanced Research Projects Agency-Advanced Plant Technologies grant HR0011-18-2-0049 (S.C.L., C.N.S., C.A.V.), and the MIT Climate Grand Challenges (CGC) program (C.A.V.). USDA Hatch funding to C.N.S. and S.C.L. is acknowledged.

## Author contributions

TT, AB, SCL, CNS and CAV conceived the study and designed the experiments. TT, AB, and AP performed the experiments and analyzed the data. QY provided technical assistance with plant genetic engineering. BF developed the arsenic sensor. TT, AB, and CAV wrote the manuscript.

## Competing interests

The authors declare no competing interests.
