## [Peer Review File · Nature Communications]

Reviewers' Comments:

Reviewer #1:

Remarks to the Author:

The manuscript by Toth et al. reports the development of a microbe-to-plant communication system using synthetic genetic circuits in both species. Leveraging a previously identified and relatively orthogonal bacterial quorum-signaling biosynthesis/perception system from *Rhodopseudomonas palustris*, *Pseudomonas putida* KT2440 "sender strains" and *Arabidopsis thaliana* "receiver lines" were engineered to secrete and perceive the signaling molecule p-coumaroyl-homoserine lactone (pC-HSL), respectively.

pC-HSL receiver *Arabidopsis* lines were constructed by transforming plants with genetic circuits that constitutively express an evolved prokaryotic RpaR repressor protein translationally fused to a VP16 activator and a nuclear localization signal. In the presence of pC-HSL, this synthetic activator could induce gene expression from a minimal 35S promoter fused to the RpaR operator sequence. The authors placed this promoter upstream of GFP, which enabled pC-HSL induction of GFP within plant cells. The transgenic circuit was initially tested in axenic 24-well plate hydroponic plant cultures supplemented with exogenous pC-HSL. GFP fluorescence was quantified by microscopy across whole root tissues and was observed to follow a Hill function-type induction pattern. This plant sensor circuit was not induced by other quorum sensing homoserine lactones and the sensor demonstrated inducibility in a soil environment.

To generate a sender *P. putida* strain, a constitutively expressed pC-HSL biosynthesis+mCherry reporter operon was assembled on a multicopy plasmid and used to transform KT2440. pC-HSL production in liquid culture and on a solid media was confirmed using an *E. coli* strain carrying an RpaR-based pC-HSL sensor circuit. TetR/aTc and LacI/IPTG induction circuits were also built on plasmids and verified to function in KT2440. To test the secrete-and-respond capability of their transkingdom sender/receiver circuits, the authors inoculated sender *Pseudomonas* with *Arabidopsis* receiver lines using their hydroponic growth assay. Fluorescence imaging showed that only sender *Pseudomonas* strains, and not wild-type *Pseudomonas* or no-bacteria controls, induced receiver *Arabidopsis*-based GFP expression. Modified sender circuits in *Pseudomonas* were constructed that induce pC-HSL production in response to either aTc or IPTG. *Pseudomonas* strains carrying these circuits were inoculated with receiver *Arabidopsis*, and Buffer and OR gate-type logic for *Arabidopsis* GFP expression was described based on the supplementation/absence of each inducer. Lastly, the authors constructed a transgenic potato line that carried the receiver circuit, although the pC-HSL-driven circuit output signal was low and *gfp* expression could only be detected using qRT-PCR. Inoculation of the constitutive sender *Pseudomonas* strain with receiver potato line demonstrated an increase in *gfp* transcript levels.

The identification of pC-HSL and its associated plant and bacterial receiver/sender parts will be of interest to the plant synthetic biology community that seeks to further apply this molecule as a transkingdom signaling modality. The receiver plant line engineering workflow also seems useful, as assaying and quantifying circuit induction under hydroponic culture appears relatively straightforward and is likely applicable to other target inducer/circuit parts. This is also the first time I have observed a Hill function-type response for a plant synthetic biology circuit. While the work is similar in form to a previous plant-to-microbe transkingdom signaling study (Geddes et al. *Nat. Commun.* 2019), the authors note that this is the first microbe-to-plant transkingdom signaling system. However, the impact of the study is limited by only a single rhizobacterial strain being tested, functional environmental signals not being used to induce bacterial circuitry, and relatively limited characterization on the spatiotemporal nature of bacterial colonization/circuit induction. The authors note that their system affords high modularity on the bacterial side, but this was not tested in a range of bacterial species. Cell type-specific induction of the receiver circuitry was also not thoroughly characterized. Additionally, bacterial induction of the plant receiver circuit was only tested in hydroponic plant culture and not more real-world applicable solid media/soil conditions. I recommend

that major revisions be performed to address some of these limitations prior to publication.

Major Comments

Lines 224-235 – While the authors have quantified bacterial pC-HSL production in the absence of plants (Figure 3), it is not clear how bacterial pC-HSL production is affected by the presence of plants (e.g., root exudates/plant factors may impact pC-HSL biosynthetic titers). Relatedly, it is not clear whether pC-HSL levels in the presence of bacteria are inducing receiver Arabidopsis circuits to the same extent as exogenous pC-HSL supplementation that was performed axenically. Arabidopsis receiver circuit sensitivity/degree of induction may differ in the presence of bacteria+pC-HSL due to bacterial effects on overall plant physiology/gene regulation (e.g., immune activation). As understanding these relationships is important to more broadly apply the authors' system and improve forward engineering of transkingdom signaling, the authors should experimentally dissect these various factors. The authors should sample the sender/receiver hydroponic system and compare how the receiver induction from bacterial-generated pC-HSL levels compares to exogenous pC-HSL supplementation. This could be accomplished by sampling the hydroponic solution, filtering to remove cells/debris, and mixing it with their receiver *E. coli* strain (as they did in Figure 3b). Additionally, the authors should inoculate hydroponic Arabidopsis cultures with wild-type *P. putida* and known concentrations of pC-HSL – this would decouple bacterial production of pC-HSL from receiver induction and enable assessment for how the presence of bacteria affects the transgenic circuit's functioning.

Line 231 – What do the author's mean by phenotypic changes? The phenotypic analysis for Arabidopsis is rather qualitative – the authors should perform similar analysis as was done in SI Fig 12 (root length, biomass measurements, etc).

Line 235 – The authors should build a 'wild-type' mCherry strain to compare colonization, does including the biosynthetic operon substantially modulate the extent of colonization and thus the extent of signal? This should also be measured by a more direct means (e.g., colony forming units).

Lines 248-254 – The authors earlier demonstrated that their engineered receiver Arabidopsis lines could perceive pC-HSL in the soil environment, but did not test whether their sender *P. putida* strain or sender strain/receiver plant system works in the soil environment. The authors should perform these experiments to further assess the field translatability of their genetic constructs.

Lines 268-271 – The authors appear to have only tested the presence or absence of the *P. putida* sender strain. As the presence of bacteria alone may affect expression of the *gfp* transgene, the authors should repeat this experiment with no bacteria, wt *Putida*, and *Putida*+sender circuit. Given that induction is about an order of magnitude lower than exogenously supplemented pC-HSL (SI Figure 13), the lower values may be due to spurious transgene induction based on the presence of bacteria/immune response.

Minor Comments

Line 64 Correction – Should be acyl-homoserine lactone, not acetyl-

Line 144 – For clarity, authors should add what cognate sensor protein senses these HSLs

Line 150-The OC12-HSL has the higher dynamic range but is not utilized further or mentioned at all, why is this?

Line 157 – Authors should measure root & shoot biomass to assess this (e.g., root length)

Line 178 – What would the effective concentration of HSL be?

Figure 2e,h – Authors should include fold-inductions in the graph

Lines 215-220 – While it is discussed in the next section of the manuscript, it's not immediately obvious why these IPTG/aTc-inducible strains are built/described. The authors should provide some context in this section to explain their development.

Reviewer #2:

Remarks to the Author:

This manuscript describes a new engineered communication system between bacteria and plants, with a longer term aim to develop this into a system in which bacteria sense environmental toxins/nutrients/other environmental cues that can then be transmitted to a plant and induce a transcriptional control in the plant to respond in an appropriate way, e.g. detoxification, induction of appropriate metabolic pathways.

The authors used a quorum sensing related signal (p-coumaroyl-homoserine lactone, pC-HSL) that certain bacteria use, but no bacteria are known to synthesise the product by themselves, instead they require the precursor from the plant host (p-coumarate). The biosynthesis pathway for full synthesis of the signal was built into a bacterium, *Pseudomonas putida*, which is a known plant growth promoting bacterium. A plant host (*Arabidopsis* and potato) was then transformed to express a sensor for the pC-HSL (based on the bacterial sensor) that was then driving GFP expression.

The authors demonstrated that 1) *Arabidopsis* plants grown in sterile liquid culture or sterile soil could produce GFP in response to added pC-HSL, and this was shown to be concentration dependent and specific to the signal and not other, related AHLs. 2) *Arabidopsis* plants grown in sterile liquid culture responded to *P. putida* that constitutively produced pC-HSL, or *P. putida* with inducible pC-HSL synthesis, with GFP production in roots. *P. putida* was colonising the root surface in those experiments, as demonstrated with an mCherry reporter in *P. putida*. 3) Potato plants transformed with the receiver construct also responded with GFP production in response to *P. putida* inoculated with *P. putida* constitutively producing pC-HSL. However, GFP production was too low for visualisation and was detected by qPCR.

Overall, the experiments were carefully designed and carried out, experimental detail is well documented, and the results are new and convincing.

My main comment would be that this is a useful prototype for gene expression driven by bacterial sensors under controlled conditions. However, its utility will depend on whether it works under real soil conditions. This is a common problem in trying to engineer plant-microbial communication, as the soil environment is complex and unpredictable. For example, non-sterile field soil could contain other signals that trigger expression of the receiver, soil pH and enzymes contained in soil could degrade the signal; the inducer bacteria might not colonise the root surface sufficiently to trigger a response in the plant.

So my main recommendation would be to test this system in unsterilised field soils and see if it still works.

Some more specific comments:

1. These sensors could be useful as sentinels, but plants can sense a lot of signals, nutrients, pathogens, toxins etc themselves and respond to them. Are there specific examples of what could be engineered to sense something that plants can't respond to where a bacterial sensor would be necessary? For example, I liked the TNT detection example, but would there be easier ways to detect TNT than engineering microbes that can sense it and then trigger a plant response? Sorry to be a bit critical here, I certainly like the whole concept, but whether it works and is useful really depends on the environmental inputs and ease of use in the field.

2. Figure 2B and G: GFP expression in the roots is a bit patchy. Were the non-expressing cells damaged somehow or why was the response not more uniform?
3. Figure 2F: this figure was shown to demonstrate that the pC-HSL receiver plants did not have a phenotypical difference to WT plants, but the plant shown looks a lot bigger (more leaves) than the WT. For potato plants a more rigorous phenotypic comparison was made, but I couldn't see it for Arabidopsis.
4. Figure 3C; Were there some photos missing here?
5. Figure 4D: GFP expression doesn't seem to be spatially correlated to where *P. putida* is colonising the roots. Is there any explanation for this?
6. When plants were induced with pC-HSL the authors used 100 micro M concentrations, this is quite high. While a concentration curve was presented, could some photos be shown of GFP expression in roots induced with 100 nM or 1 microM concentrations as a comparison? That would be a more realistic concentration in the soil.
7. Line 64: change 'acetyl-homoserine' to 'acyl-homoserine'
8. Line 78: what does 'fewer part libraries' refer to?
9. Line 235: 'wild type bacteria adhered similarly': How was this determined if they didn't express mCherry? Were they visualised in a different way?
10. Line 337: What were the DMF stocks of HSLs diluted with? Was the pH adjusted (the lactone ring is pH sensitive)?
11. Line 405: the first sentence is unfinished. Also, what is an HTS attachment?
12. Line 416: change to 'values were quantified'.
13. The references will need some corrections in the end, i.e. make sure journal names are always capitalised, species names are italicised etc.

Reviewer #3:

Remarks to the Author:

The Toth et al manuscript aims to develop an inter-kingdom communication channel between a beneficial bacterium and Arabidopsis plants. The manuscript develops potentially exciting tools that can be used to engineer bacteria and plants and create a bacterial sentinel capable of relaying information to the plant in a changing environment. Although the idea of coupling bacterial sensors to plants is not new, the tools developed in this work could be interesting because they have minimal interference with the complex microbial background present in the soil.

However, the data presented does not fully support the conclusions of the manuscript. I have detected important experimental design problems, lack of key information, inappropriate use of technical language, and organizational problems in the manuscript. Therefore, I consider that this manuscript is not of sufficient quality in its current version to be accepted for publication in Nature Comm.

Comments:

- 1- The concentration used to induce the HSL system in vitro experiments is very high, out of the physiological range for the inducer.
- 2- Line 150. Arabidopsis lines containing the pC-HSL and OC12-HSL receivers showed 6-fold and 40-fold. Where is this information in the manuscript?
- 3- Supplementary Figure 5 is cited before Supplementary Figure 3 and 4.
- 4- For the OHC14-HSL receiver, we only found a line that produced a low 2-fold dynamic range and the OC6-HSL receiver yielded no functional lines, so neither were pursued further. This information is not in the manuscript.
- 5- nine independent lines tested for pC-HSL induction, eight were active, of which we selected *A. thaliana* 315_14_5 for further characterization. This data is not present in the manuscript.
- 6- We observed higher levels of fluorescence in mature root tissue likely because those cells have had more time to express the receiver, respond to the molecule, and express GFP. This conclusion is a speculation. Where is this data?

- 7- Whole plant imaging showed induction of tissue throughout the plant, with higher induction in the roots (Figure 2a). This information is not present in this figure. The authors have to quantify GFP intensity in plant organs to conclude this. At the moment it is just speculative.
- 8- No changes were observed in growth or root morphology between wild-type *A. thaliana* and *A. thaliana* 315_14_5 (Figure 2c,f). To conclude this the plant phenotypes should be quantified. The figure calls are not in order. Panel f appears before e.
- 9- The full response functions were measured for the pC-HSL and OC12-HSL receivers (Figure 2d, Supplementary Figure 5). Which panel in Fig 5 contains this information?
- 10- Line 162. This equation should be moved to M&M section.
- 11- The minimum detection limit is 100 nM pC-HSL, which is approximately an order of magnitude higher than the detection limit of RpaR in *R. palustris*. In the absence of inducer, the background expression was similar to wild-type *A. thaliana*. Where I can find this information in the manuscript?
- 12- Plants containing the pC-HSL receiver were then tested for orthogonality in responding to other HSLs (Figure 2e). What was the concentration used? How the authors know the sensitivity to these molecules? Why only one concentration used?
- 13- After the emergence of the two cotyledons and the first leaf. It obvious that the first leaf appears after the two cotyledons. It is not necessary to mention the cotyledons.
- 14- growth chamber before being induced in situ (Methods). What in situ induction means in this context?
- 15- by pipetting 1 mL of water supplemented with 100 μ M of pC-HSL directly at the plant-soil interface. What is the final concentration of the ligand. This is not an accurate way to mention the concentration of a ligand.
- 16- Plants were then grown for an additional 24 hours in the growth chamber before being prepared for imaging by washing the roots in water. GFP fluorescence was captured using confocal microscopy (Methods). Similarly, to the hydroponics experiments, GFP fluorescence in the root tissue of the pC-HSL receiver was activated by the presence of pC-HSL in the soil. This is not a canonical way to write the results section, it looks like M&M.
- 17- Further, it has been reported to not contain HSL-producing enzymes (unlike *P. putida* IsoF and WCS358, which produce 3OC12-HSL), nor known pathways to HSL mimics, which we confirmed through genome analysis (Methods). This conclusion needs a figure.
- 18- Line 197 under the control of a strong constitutive promoter. Which promoter?
- 19- There are supplementary figures that are cited only in others figures legends.
- 20- From these data, the concentration produced by the *P. putida* sender was estimated to be 70 ± 10 nM pC-HSL. Then, the impact of adding 100 μ M p-coumarate to the media was tested because it has been shown to increase pC-HSL production. Indeed, this led to an increase in pC-HSL production to 240 ± 40 nM. This concentration is sufficient for inducing the *Arabidopsis* pC-HSL receiver. Where I can find this data?
- 21- Several figure panels are cartoons not real data.
- 22- *A. thaliana* by mixing 25% LB media with 75% MS media (Methods). The plants were germinated on solid agar and added to the 24-well plates. Separately, the bacteria were grown in LB media overnight and then diluted to a starting OD600 of 0.01 into the wells containing the plants. The co-culture was grown for 24 hours and then the roots imaged. It is not necessary to mix LB with MS, the majority of the bacteria can survive in MS medium when the plant is present. This is not a good experimental design. With this experimental design it is impossible to discriminate between the ligand that is secreted and accumulates in the medium from the ligand that is produced in the vicinity of the root or in the root. This experimental design is equivalent to adding the pure compound to the medium. Therefore, despite the reduced concentration of the ligand produced by the bacteria, the plant is able to induce *gfp*.
- 23- observable phenotypic changes as compared to the plants grown without the bacteria (Figure 4b). Quantification of the plant phenotypes is needed.

All in all, I think that this manuscript has major experimental design problems and has no quality enough to be published in Nature Comm.

Reviewer #4:

Remarks to the Author:

The manuscript describes engineered quorum sensing communication between bacteria (*P. putida*) and plants (*A. thaliana* and *S. tuberosum*). This involves the implementation of sender modules in bacteria and the corresponding receiver modules in plants. Experiments characterise various systems, more in detail the pC-HSL one, which was the most promising. Authors carry out several tests, with bacteria-bacteria, plants in wells, plants in soil... to cover a wide range dynamics and controls. They also engineered regulated versions of the communication channel (with IPTG and aTc) for testing sensing applications. The conceptual idea is simple (something positive to highlight) and the experimental setup convincing.

A synthetic biology approach to plant-microbe interactions is something novel that deserves more attention. In my opinion, this manuscript contributes towards that goal by providing a tool that would be of use to the community. The use of soil bacteria enhances the work done.

Just a few minor comments on questions I had while reading the manuscript.

- Not sure how plants growing in soil (Figure 2f) were characterised. If I get it right, the signal was inoculated on soil directly. What root section was measured? Was all the root induced? To me, an overarching challenge would be to make the communication channel effective in a real-life scenario, since cells will not be homogeneously distributed, etc. Maybe a mention to this in the discussion—as a future challenge—would be welcome.
- Adding p-coumarate increased decisively the production of pC-HSL. Does that mean that the communication channel only works when p-coumarate is present? If so, does this imposes a constraint to the system?
- It is hard to interpret Figure 4f as an OR logic. I understand the reference, of course. But since the error is too wide, I would find analogue terms more appropriate.
- Bacteria were grown on plates with 0.8% agar. This means bacteria would swim to fill the entire space, right? Did authors observe any root colonisation dynamics?

Reviewers' comments:

Reviewer #1:

Note that the lettered comments came from the "intro"

A. *The impact of the study is limited by only a single rhizobacterial strain being tested. The authors note that their system affords high modularity on the bacterial side, but this was not tested in a range of bacterial species.*

In the original submission, the communication system was tested in *Pseudomonas putida* KT2440. In the revision, we have now also demonstrated that *Klebsiella pneumoniae* 342, a nitrogen-fixing strain. New data include communication with the root via the constitutive expression of pC-HSL (Figure 3c,d) and where pC-HSL is under the control of an arsenic sensor (Figure 4b).

Note that by "modularity," we intended to refer to the ability of the communication system to be connected to different sensors or circuits. The claim is not that the genetic devices can be easily moved between species. We had edited the text (Introduction and caption of Figure 1) to clarify this point.

B. *Functional environmental signals not being used to induce bacterial circuitry*

We now show that we can engineer *P. putida* and *K. pneumoniae* to sense arsenic and communicate this information to the plant roots (Figure 4b). Note that the EPA arsenic standard for drinking water is 10 ppb and the limit of detection of our *P. putida* arsenic sensor is about 10 ppb. Toxic heavy metals in farmland is a critical problem.

C. *Relatively limited characterization on the spatiotemporal nature of bacterial colonization/circuit induction. Cell type-specific induction of the receiver circuitry was also not thoroughly characterized.*

We have included new experiments showing the spatiotemporal characterization of the activation of the receiver device in *A. thaliana* roots (Figure 2a, Supplementary Figures 6 and 7). We found that GFP was only expressed in mature tissues, with no GFP in the meristem. In the maturing tissue, close to the meristem, we could observe GFP in the nucleus. In mature tissues, especially the epidermis, GFP expression was both in the nucleus and in the membrane. There was no GFP expression in the root hairs.

The species of bacteria we are using do not colonize *A. thaliana* roots. This is consistent with the literature, noting that colonization can occur after longer experimental times (>9 days) (Wu, Comparative genomics and functional analysis of niche-specific adaptation in *Pseudomonas putida*. *FEMS Microbiology Reviews*, 2011; Dong, Quantitative assessments of the host range and strain specificity of endophytic colonization by *Klebsiella pneumoniae* 342, *Plant and Soil*, 2003).

1. *Lines 224-235 – While the authors have quantified bacterial pC-HSL production in the absence of plants (Figure 3), it is not clear how bacterial pC-HSL production is affected by the presence of plants (e.g., root exudates/plant factors may impact pC-HSL biosynthetic titers). Relatedly, it is not clear whether pC-HSL levels in the presence of bacteria are inducing receiver Arabidopsis circuits to the same extent as exogenous pC-HSL supplementation that was performed axenically. Arabidopsis receiver circuit sensitivity/degree of induction may differ in the presence of bacteria+pC-HSL due to bacterial effects on overall plant physiology/gene regulation (e.g., immune activation). As understanding these relationships*

is important to more broadly apply the authors' system and improve forward engineering of transkingdom signaling, the authors should experimentally dissect these various factors. The authors should sample the sender/receiver hydroponic system and compare how the receiver induction from bacterial-generated pC-HSL levels compares to exogenous pC-HSL supplementation. This could be accomplished by sampling the hydroponic solution, filtering to remove cells/debris, and mixing it with their receiver E. coli strain (as they did in Figure 3b). Additionally, the authors should inoculate hydroponic Arabidopsis cultures with wild-type P. putida and known concentrations of pC-HSL – this would decouple bacterial production of pC-HSL from receiver induction and enable assessment for how the presence of bacteria affects the transgenic circuit's functioning.

The two experiments suggested by the reviewer have been performed to demonstrate that the plant has minimal effect bacterial production of pC-HSL. These experiments are described below and are summarized in the main text, with data in the SI.

First, an experiment was performed to determine how pC-HSL production by the bacterium is affected by the presence of the plant. pC-HSL producing bacteria were grown with inducer in the presence and absence of the plant (Supplementary Figure 22a). After the growth, the cultures were spun down and filtered and the pC-HSL concentration determined, as done previously. Induction still occurs in the presence or absence of the plant as expected, although there is some lowering of the pC-HSL concentration (2.34 ± 0.05 versus 1.54 ± 0.21 μM), which could be due to sequestration by the plant.

Second, an experiment was performed to determine if the presence of the bacteria themselves influence the induction of the pC-HSL receiver. The *A. thaliana* pC-HSL receiver was induced with pC-HSL in the presence and absence of WT bacteria (Supplementary Figure 22b). The presence of WT bacteria had a minimal effect on the pC-HSL receiver response function.

2. Line 231 – What do the authors mean by phenotypic changes? The phenotypic analysis for Arabidopsis is rather qualitative – the authors should perform similar analysis as was done in SI Fig 12 (root length, biomass measurements, etc).

We have performed phenotypic analysis comparing *A. thaliana* wild-type and *A. thaliana* with the pC-HSL receiver (Figure 2e). The differences are not significant.

3. Line 235 – The authors should build a 'wild-type' mCherry strain to compare colonization, does including the biosynthetic operon substantially modulate the extent of colonization and thus the extent of signal? This should also be measured by a more direct means (e.g., colony forming units).

These strains do not colonize plant roots (see Comment C). We have not observed any difference in growth or presence around the root due to carrying the biosynthetic genes, but this is difficult to quantify.

4. Lines 248-254 – The authors earlier demonstrated that their engineered receiver Arabidopsis lines could perceive pC-HSL in the soil environment but did not test whether their sender P. putida strain or sender strain/receiver plant system works in the soil environment. The authors should perform these experiments to further assess the field translatability of their genetic constructs.

We have now performed soil experiments (Figure 3e,f). *P. putida* was able to activate GFP expression in the plant pC-HSL receiver in sterile soil and non-sterile soil.

5. *Lines 268-271 – The authors appear to have only tested the presence or absence of the P. putida sender strain. As the presence of bacteria alone may affect expression of the gfp transgene, the authors should repeat this experiment with no bacteria, wt Putida, and Putida+sender circuit. Given that induction is about an order of magnitude lower than exogenously supplemented pC-HSL (SI Figure 13), the lower values may be due to spurious transgene induction based on the presence of bacteria/immune response.*

We have performed qRT-PCR experiments to quantify the induction of the pC-HSL receiver (through its gfp output) with the suggested combination of conditions (Supplementary Figure 15). The background inductions of the receiver are similar when compared between 0 mM pC-HSL and the presence of wild-type P. putida. The induction by 1 μ M pC-HSL and the bacteria producing pC-HSL are also similar, noting that it is not an exact comparison because of variability in pC-HSL production by the bacteria.

6. *Line 64 Correction – Should be acyl-homoserine lactone, not acetyl-*

This has been corrected.

7. *Line 144 – For clarity, authors should add what cognate sensor protein senses these HSLs*

We have edited the text to clarify which regulators bind to which HSL.

8. *Line 150-The OC12-HSL has the higher dynamic range but is not utilized further or mentioned at all, why is this?*

This sentence is a typo. The pC-HSL receiver has a higher dynamic range; we have corrected the manuscript and edited the figure caption.

9. *Line 157 – Authors should measure root & shoot biomass to assess this (e.g., root length)*

We have made these measurements and the data are included in Figure 2e. These data do not change the conclusion.

10. *Line 178 – What would the effective concentration of HSL be?*

It is very difficult to estimate the effective soil concentration would be. Based on how much we add, the effective concentration would be about 0.1 μ M at root depth in these experiments (Methods).

11. *Figure 2e,h – Authors should include fold-inductions in the graph*

The fold-inductions have been added to the caption.

12. *Lines 215-220 – While it is discussed in the next section of the manuscript, it's not immediately obvious why these IPTG/aTc-inducible strains are built/described. The authors should provide some context in this section to explain their development.*

The text has been edited to provide context. Note that these are common inducible systems intended to evaluate the ability of the bacteria to process multiple signals using genetic circuits (logic operations).

There is not an agricultural reason to use these sensors per se. However, we have included an additional arsenic sensor to expand into relevant agricultural/environmental sensors.

Reviewer #2:

1. *My main comment would be that this is a useful prototype for gene expression driven by bacterial sensors under controlled conditions. However, its utility will depend on whether it works under real soil conditions. This is a common problem in trying to engineer plant-microbial communication, as the soil environment is complex and unpredictable. For example, non-sterile field soil could contain other signals that trigger expression of the receiver, soil pH and enzymes contained in soil could degrade the signal; the inducer bacteria might not colonise the root surface sufficiently to trigger a response in the plant. So my main recommendation would be to test this system in unsterilised field soils and see if it still works.*

Soil experiments have been performed and the data are provided in Figure 3e-f and Supplementary Figure 23-24. These experiments were performed in sterile soil and non-sterile soil. The *A. thaliana* pC-HSL receiver was inoculated either by watering or by seed soaking. Whole-root imaging of the plants inoculated by watering in sterile and non-sterile soil show that GFP is expressed in the mature root tissues close to the soil-air interface (Supplementary Figure 24).

2. *These sensors could be useful as sentinels, but plants can sense a lot of signals, nutrients, pathogens, toxins etc themselves and respond to them. Are there specific examples of what could be engineered to sense something that plants can't respond to an where a bacterial sensor would be necessary? For example, I liked the TNT detection example, but would there be easier ways to detect TNT than engineering microbes that can sense it and then trigger a plant response? Sorry to be a bit critical here, I certainly like the whole concept, but whether it works and is useful really depends on the environmental inputs and ease of use in the field.*

There are far fewer synthetic sensors available for plants. Even inducible systems are notoriously unavailable for plants and the few that are used are far worse in their performance and dynamic range than what is available for bacteria. In contrast, there are hundreds of sensors available for bacteria. In addition, the genetic optimization of a sensor is much easier in bacteria because it is possible to implement directed evolution techniques easily and rapid growth means faster design-build-test cycles. Plants grow slowly and few variants can be tested, leading to poorly performing sensors. Also, the accessibility of a molecule to different plant tissues may differ. The molecule needs to be taken up in the root, transported, and cross the cell and nuclear membranes. Finally, it is possible to build complex genetic circuits in bacteria that can integrate multiple signals or produce a dynamic response.

The plant receiver allows one plant-based sensor to be developed and optimized. It can then be used with all the sensors and circuits available for bacteria. Critically, changing what is being sensed and the signal processing involves changing the bacterium to which the plant is coupled and not building a new system in the plant. Also, by using a HSL, transport into the plant and across cell membranes are simplified and standardized. Thus, the signal can be received outside of the root and then transmitted into the plant into the plant as a separate step.

2. *Figure 2B and G: GFP expression in the roots is a bit patchy. Were the non-expressing cells damaged somehow or why was the response not more uniform?*

We have included new experiments showing the spatiotemporal characterization of the activation of the receiver device in *A. thaliana* roots (Figure 2a, Supplementary Figure 6-7). We found that the “patchiness” of the images was due to the plane in which the confocal images were taken. GFP expression was observed in the nucleus and the membrane of mature tissues and there was no GFP in the root hairs. Therefore, depending on the confocal plane, the nucleus was more or less apparent,

giving the impression of “patchiness” as GFP expression was not uniform across the cell. Using PI staining, we confirmed that there was no damage to the cells with lower fluorescence (membranes were intact).

3. *Figure 2F: this figure was shown to demonstrate that the pC-HSL receiver plants did not have a phenotypical difference to WT plants, but the plant shown looks a lot bigger (more leaves) than the WT. For potato plants a more rigorous phenotypic comparison was made, but I couldn't see it for Arabidopsis.*

New experiments have been performed to quantify the phenotypic traits of *A. thaliana* wild-type and carrying the pC-HSL receiver (Figure 2e). These are the same measurements as were provided for *S. tuberosum* in the original submission. Upon quantification, there are no statistically significant differences between the phenotypes whether *A. thaliana* carries the pC-HSL receiver or not.

4. *Figure 3C; Were there some photos missing here?*

We have now fixed the issue (sometimes the images would not show).

5. *Figure 4D: GFP expression doesn't seem to be spatially correlated to where P. putida is colonizing the roots. Is there any explanation for this?*

We found that in the presence of the *A. thaliana* pC-HSL receiver, *P. putida* pTT337 produces 1.5 ± 0.2 μM pC-HSL (Supplementary Figure 22). We expect the concentration in the hydroponic system to be homogeneous and therefore the GFP expression should not be correlated (Figure 2c, Supplementary Figure 12).

6. *When plants were induced with pC-HSL the authors used 100 uM concentrations, this is quite high. While a concentration curve was presented, could some photos be shown of GFP expression in roots induced with 100 nM or 1 uM concentrations as a comparison? That would be a more realistic concentration in the soil.*

We had measured the response to different concentrations of pC-HSL. We now show the images from 1 μM pC-HSL (Figure 2a, Supplementary Figure 10 and 12) and 100 nM pC-HSL (Supplementary Figure 10 and 12).

7. *Line 64: change 'acetyl-homoserine' to 'acyl-homoserine'*

The typo has been corrected.

8. *Line 78: what does 'fewer part libraries' refer to?*

The text has been edited for clarity.

9. *Line 235: 'wild type bacteria adhered similarly': How was this determined if they didn't express mCherry? Were they visualized in a different way?*

We removed this statement from the main text. (We had observed it using brightfield but had not quantified it sufficiently to support this statement.)

10. *Line 337: What were the DMF stocks of HSLs diluted with? Was the pH adjusted (the lactone ring is pH sensitive)?*

The DMF stocks were diluted into ddH₂O before used to induce the bacteria-plant system. The pH of the plant medium was adjusted to 5.7 using KOH before adding the inducer. Upon adding the precursor to MS medium, the pH is 4.6 (final concentration 100 μ M p-coumarate). HSL ring breakage occurs under alkaline conditions.

11. *Line 405: the first sentence is unfinished. Also, what is an HTS attachment?*

The sentence has been edited and HTS defined. High Throughput Sampler (HTS) refers to the autosampler of the flow cytometer, indicating that the samples were run in 96-well plates rather than in tubes.

12. *Line 416: change to 'values were quantified'.*

This change has been made.

13. *The references will need some corrections in the end, i.e. make sure journal names are always capitalised, species names are italicised etc.*

The references have been edited.

Reviewer #3:

A. *Although the idea of coupling bacterial sensors to plants is not new, the tools developed in this work could be interesting because they have minimal interference with the complex microbial background present in the soil.*

We are unaware of any prior literature to couple bacterial sensors to plants. In terms of putting bacterial sensors into plants, then yes there has been a lot of work in this space (on the backs of which we constructed the pC-HSL receiver). However, there has not been prior work to have the bacteria sense a signal and then communicate this information to the plant.

1. *The concentration used to induce the HSL system in vitro experiments is very high, out of the physiological range for the inducer.*

We had measured the respond to different concentrations of pC-HSL. We now show the images from 1 μ M pC-HSL (Figure 2a, Supplementary Figure 10 and 12) and 100 nM pC-HSL (Supplementary Figure 10 and 12).

2. *Line 150. Arabidopsis lines containing the pC-HSL and OC12-HSL receivers showed 6-fold and 40-fold. Where is this information in the manuscript?*

This was a typo; we have corrected the manuscript and edited the figure caption.

3. *Supplementary Figure 5 is cited before Supplementary Figure 3 and 4.*

This has been corrected.

4. *For the OHC14-HSL receiver, we only found a line that produced a low 2-fold dynamic range and the OC6-HSL receiver yielded no functional lines, so neither were pursued further. This information is not in the manuscript.*

We have included the data for the other receiver lines, including the OHC14-HSL and the OC6-HSL receivers (Supplementary Figure 4).

5. *Nine independent lines tested for pC-HSL induction, eight were active, of which we selected *A. thaliana* 315_14_5 for further characterization. This data is not present in the manuscript.*

We have included data for the induction with either 0 μ M or 100 μ M of pC-HSL for eight independent lines (Supplementary Figure 5). Out of these, six showed a response to the chemical induction with pC-HSL. The ninth line is the one used Figure 2b-d (*A. thaliana* 315_14_5).

6. *We observed higher levels of fluorescence in mature root tissue likely because those cells have had more time to express the receiver, respond to the molecule, and express GFP. This conclusion is a speculation. Where is this data?*

We have added additional figures to support this claim (Figure 2a and Supplementary Figures 6,7).

7. *Whole plant imaging showed induction of tissue throughout the plant, with higher induction in the*

roots (Figure 2a). This information is not present in this figure. The authors have to quantify GFP intensity in plant organs to conclude this. At the moment it is just speculative.

To support this claim, we have quantified GFP transcripts using RT-qPCR to show that GFP induction is higher in the roots than in leaf tissues (Supplementary Figure 9).

8. *No changes were observed in growth or root morphology between wild-type A. thaliana and A. thaliana 315_14_5 (Figure 2c,f). To conclude this, the plant phenotypes should be quantified. The figure calls are not in order. Panel f appears before e.*

New experiments have been performed to quantitatively compare the phenotypes of wild-type A. thaliana and A. thaliana 315_14_5 (Figure 2e). We did not find statistically significant differences between the phenotypes of the lines.

9. *The full response functions were measured for the pC-HSL and OC12-HSL receivers (Figure 2d, Supplementary Figure 5). Which panel in Fig 5 contains this information?*

The panel is Supplementary Figure 3 for the OC12-HSL receiver and Figure 2c for pC-HSL.

10. *Line 162. This equation should be moved to M&M section.*

We have kept this equation in the main text; we prefer to have the math shown there and note it is rare to provide plant circuit induction data in this quantitative context.

11. *The minimum detection limit is 100 nM pC-HSL, which is approximately an order of magnitude higher than the detection limit of RpaR in R. palustris. In the absence of inducer, the background expression was similar to wild-type A. thaliana. Where I can find this information in the manuscript?*

We found that the background expression of the A. thaliana pC-HSL receiver was 7-fold higher than that of the wild-type A. thaliana. This information is now provided in Supplementary Figure 11 and we have added a description of it to the main text.

12. *Plants containing the pC-HSL receiver were then tested for orthogonality in responding to other HSLs (Figure 2e). What was the concentration used? How do the authors know the sensitivity to these molecules? Why only one concentration used?*

The figure caption has been edited to include the concentrations (100 μ M for each inducer). These concentrations were selected because they represent the maximum concentrations used to induce these sensors. Since we are looking for crosstalk in these experiments, the highest concentration is appropriate and the standard in field. Note that we do now include additional data and images for lower concentrations (Figure 2a, Supplementary Figures 10 and 12), but the highest induction is still used to test for orthogonality.

13. *After the emergence of the two cotyledons and the first leaf. It obvious that the first leaf appears after the two cotyledons. It is not necessary to mention the cotyledons.*

This comment has been deleted from the text, as suggested.

14. *Growth chamber before being induced in situ (Methods). What in situ induction means in this context?*

***In situ* referred to experiments being performed in the soil. The term was intended to highlight that the plant was not transferred to a soil pre-mixed with the inducer. Rather, the inducer was added to the soil in which the plant had already been growing. The text has been edited for clarity.**

15. *By pipetting 1 mL of water supplemented with 100 μ M of pC-HSL directly at the plant-soil interface. What is the final concentration of the ligand. This is not an accurate way to mention the concentration of a ligand.*

We estimate the concentration of pC-HSL in the soil to be 0.1 μ M in the region of the root. The text has been edited for clarity and this calculation appears in the Methods.

16. *Plants were then grown for an additional 24 hours in the growth chamber before being prepared for imaging by washing the roots in water. GFP fluorescence was captured using confocal microscopy (Methods). Similarly, to the hydroponics experiments, GFP fluorescence in the root tissue of the pC-HSL receiver was activated by the presence of pC-HSL in the soil. This is not a canonical way to write the results section, it looks like M&M.*

We have edited the results.

17. *Further, it has been reported to not contain HSL-producing enzymes (unlike *P. putida* IsoF and WCS358, which produce 3OC12-HSL), nor known pathways to HSL mimics, which we confirmed through genome analysis (Methods). This conclusion needs a figure.*

A BLAST search returns no hits. It is unclear how to convert this result into a figure. We have included the details of the search in the Methods so that it could be repeated.

18. *Line 197 under the control of a strong constitutive promoter. Which promoter?*

This information is now provided. The name of the promoter is BBa_J23100.

19. *There are supplementary figures that are cited only in others figures [captions].*

We believe that it is ok to cite a SI figure in the caption, but will fix it if consistent with Nature Communications guidelines.

20. *From these data, the concentration produced by the *P. putida* sender was estimated to be 70 ± 10 nM pC-HSL. Then, the impact of adding 100 μ M p-coumarate to the media was tested because it has been shown to increase pC-HSL production. Indeed, this led to an increase in pC-HSL production to 240 ± 40 nM. This concentration is sufficient for inducing the *Arabidopsis* pC-HSL receiver. Where I can find this data?*

The data is provided in Supplementary Figure 17. We have added the call to this figure in the text.

21. *Several figure panels are cartoons not real data.*

We are not sure what is being requested. Descriptive diagrams are generally acceptable figure panels.

22. *A. thaliana* by mixing 25% LB media with 75% MS media (Methods). The plants were germinated on solid agar and added to the 24-well plates. Separately, the bacteria were grown in LB media overnight and then diluted to a starting OD600 of 0.01 into the wells containing the plants. The co-culture was grown for 24 hours and then the roots imaged. It is not necessary to mix LB with MS, the majority of the bacteria can survive in MS medium when the plant is present. This is not a good experimental design. With this experimental design it is impossible to discriminate between the ligand that is secreted and accumulates in the medium from the ligand that is produced in the vicinity of the root or in the root. This experimental design is equivalent to adding the pure compound to the medium. Therefore, despite the reduced concentration of the ligand produced by the bacteria, the plant is able to induce *gfp*.

We have changed our experimental design as suggested and showed that the system works in 100% MS after a wash step to remove any LB medium or pC-HSL produced by the overnight *P. putida* or *K. pneumoniae* culture (Figures 3 and 4).

23. *Observable phenotypic changes as compared to the plants grown without the bacteria (Figure 4b). Quantification of the plant phenotypes is needed.*

We found the phenotype change difficult to quantify as the plants are only grown with the bacteria in the hydroponic system for 24 hours. This duration does not provide a sufficient time for additional growth and development; thus we have removed this claim from the main text.

Reviewer #4:

1. Not sure how plants growing in soil (Figure 2f) were characterized. If I get it right, the signal was inoculated on soil directly. What root section was measured? Was all the root induced? To me, an overarching challenge would be to make the communication channel effective in a real-life scenario, since cells will not be homogeneously distributed, etc. Maybe a mention to this in the discussion—as a future challenge—would be welcome.

In Figure 2f, pC-HSL was inoculated directly into the soil. We measured the mature root tissue located close to the surface. In this scenario, most of the mature root tissues were induced. We have now included sterile and non-sterile soil experiments of the bacteria inducing the *A. thaliana* pC-HSL receiver to demonstrate that the system functions in more realistic scenario (Figure 3e-f, Supplementary Figure 23-24). Whole-root imaging of the plants grown in sterile and non-sterile soil and inoculated by watering show that the roots closer to the surface are express the most GFP (Supplementary Figure 24). This is likely due to the cells not being homogeneously distributed, but also because GFP is expressed in mature tissues (Figure 2a, Supplementary Figures 6-7). We have added this point to the discussion.

2. Adding p-coumarate increased decisively the production of pC-HSL. Does that mean that the communication channel only works when p-coumarate is present? If so, does this imposes a constraint to the system?

No, the system also works in the absence of p-coumarate. These data are shown in Supplementary Figure 20. The addition of p-coumarate increases the signal: we observed a 11-fold induction without p-coumarate and 15-fold with p-coumarate. In terms of being a constraint, it is possible to get the same result through metabolic engineering, which has been done in *E. coli* (Du et al. 2020) and would likely be possible in soil bacteria.

(Du, P., Zhao, H., Zhang, H. et al. De novo design of an intercellular signaling toolbox for multi-channel cell–cell communication and biological computation. *Nat Commun* 11, 4226 (2020). <https://doi.org/10.1038/s41467-020-17993-w>)

3. It is hard to interpret Figure 4f as an OR logic. I understand the reference, of course. But since the error is too wide, I would find analogue terms more appropriate.

We have edited the text to refer to this as a “fuzzy OR gate.”

4. Bacteria were grown on plates with 0.8% agar. This means bacteria would swim to fill the entire space, right? Did authors observe any root colonisation dynamics?

The plates were just to obtain colonies and the bacteria do not fill the entire space. The induction experiments were performed in a hydroponic system, MS agar plates, or soil and in all cases the bacteria are well mixed.

Reviewers' Comments:

Reviewer #1:

Remarks to the Author:

The manuscript by Boo & Toth et al. has been substantially improved through revision experiments. The authors have done a nice job addressing each of the reviewer's concerns. However, a few minor comments below should first be addressed to strengthen the results and fix text/figure issues.

Major Comment:

All Figures: Statistical analysis is currently lacking for many main text/SI figures (e.g., Figs 2g, 3c, 4a-d) and should be performed. While statistical analysis for some graphs is mentioned in the figure captions, the authors should display significance/p-values within graphs when appropriate.

Minor Comments:

Line 160: The authors state "GFP expression was both in the nucleus and membrane". While GFP expression in root cell nuclei is observed, "membrane expression" may instead be a result of GFP in cytosolic space compressed against the membrane by the central cell vacuole. Without more detailed localization analysis, the authors should generalize this description.

Lines 213-230: The authors mention *Klebsiella* in preceding and later sections, but in these paragraphs the authors only characterize pC-HSL production by *Pseudomonas* and not *Klebsiella*. The authors should perform similar experiments to SI Figure 17 with *Klebsiella* to better understand its pC-HSL production titers and ability to induce 'receiver' responses, relative to *Pseudomonas*.

Lines 300-304: The authors should state in the main text what concentration of arsenic was added in the plant environment.

Figure 1a: Image of root in maturation zone is upside down. Root hairs form on the root-ward side of epidermal cells and it is clear that this image was inadvertently flipped.

Figure 2e: "Weight" is spelled incorrectly in the y-axis of two graphs.

Figure 3a/3b: The authors should mention *Klebsiella* in the Figure 3a cartoon and in Figure 3b should show a picture of the *Arabidopsis*-containing *Klebsiella*.

Figure 4b: The cartoon incorrectly shows IPTG/aTc inducing the arsenic sensor.

Reviewer #2:

Remarks to the Author:

The revised version of the manuscript is much improved and the authors have added additional experiments that addressed my previous comments. It was good to see that the response was still detectable in non-sterile soil, although it was much weaker than in sterile soil. In general, I think a word of caution should be added to the discussion about this. I am convinced that these reporter systems work well in closed systems, but once there is competition with other bacteria, possible signal breakdown or variable soil conditions in the field, the reliability of the sensors would be reduced. I am also not really convinced about examples of using this system to detect land mines. I know it's just a suggestion, but in reality, how would one first plant transgenic seeds (considering GM legislation) across large areas of potential landmine contaminated land, then harvest each plant, record the exact location, look for quantitative changes in fluorescence and then go back to the spot and decontaminate the mine. Yes, possible, maybe with drones, but not very practical.

Some minor changes:

Line 77: check spelling of Azorhizobium caulidurans

Line 135: please correct to 'leading to GFP..'

Line 144: please correct this sentence: '..allowed the plant roots to be exposed the plant roots to a homogeneous.'

Line 160: In mature tissues GFP expression was both in the nucleus and the membrane': First, please refer here to GFP localisation, not expression, Second, it is highly unlikely that GFP would localise in the membrane. This looks like cytoplasmic localisation. The cytoplasm of these cells would be a thin strip pressed against the membrane.

Line 161: There was no GFP expression in the root hairs. Do you have an explanation for this observation?

Line 163: '..showed induction of tissue throughout the root system'. Please rephrase to 'showed induction of GFP expression throughout different root tissues..' or similar.

Line 365: please change 'cinnomoyl' to 'cinnamoyl'

Line 465: Why was the UV laser used for GFP visualisation on the SP8? At ~365 nm excitation? Line

466: Please also give excitation wavelength of the argon laser.

Figures:

For all figures with histograms, please show statistical differences in the figures, not just the legends.

Figure 2: Thank you for adding phenotyping results for Arabidopsis. Did you record any root phenotypes since the highest expression of the reporter was in the root system?

Reviewer #3:

Remarks to the Author:

I would like to thank the authors for addressing some of my comments. Although some new experiments have been added to the manuscript I don't see a significant improvement in the quality of this work. I am still detecting significant mistakes in the manuscript and a lack of quality in the science described that make the manuscript not suitable for publication in this journal.

For example:

1- *A. thaliana* lines containing the pC-HSL and OC12-HSL receivers showed 40-fold and 6-fold inductions, respectively (Figure 2a-c and Supplementary Figure 3). For the OHC14-HSL receiver, we only found a line that produced a low 2-fold dynamic range (Supplementary Figure 4) and the OC6-HSL receiver yielded no functional lines, so neither were pursued further. I can't find the information about the fold changes in the manuscript.

2- In mature tissues, GFP expression was both in the nucleus and in the membrane. This information is not in the manuscript.

3- There was no GFP expression in the root hairs (Figure 2a, Supplementary Figures 6-7). The quality of the pictures and the level of details is not enough to conclude this.

4- No phenotypic differences were observed between wild-type *A. thaliana* and *A. thaliana* (Figure 2b and Figure 2e). Knowing that the expression of the reporter is root specific, no phenotypes related with root architecture were quantified such as LR density, primary root length, etc.

5- No quantitative data is provided in Supl fig 15 and 16.

6- From these data, the concentration produced by the *P. putida* sender was estimated to be 70 ± 10 nM pC-HSL. This data is not in the manuscript.

7- The co-culture was grown for 24 hours and then the roots imaged. Plants grown for this length of time with bacteria did not show observable phenotypic changes as compared to the plants grown without the bacteria (Figure 3b). Phenotypes were not quantified.

8- uninduced when grown with wild-type bacteria (Figures 3c-d, Supplementary Figures 15-16, 19). The presence of 100 μ M p-coumarate in the medium increased the response of the receiver to *P. putida* pTT337 from 11- to 15-fold (Supplementary Figure 20). Where I can see this induction?

9- The new soil experiments are open making impossible the establishment of causality between the

bacterium adding and the plant signal.

Reviewer #4:

Remarks to the Author:

I found author response to comments appropriate. All my concerns were addressed.

Reviewer #1:

1. *All Figures: Statistical analysis is currently lacking for many main text/Sl figures (e.g., Figs 2g, 3c, 4a-d) and should be performed. While statistical analysis for some graphs is mentioned in the figure captions, the authors should display significance/p-values within graphs when appropriate.*

We have edited the main text and Sl figures to show the significance of all plant experiments.

2. *Line 160: The authors state “GFP expression was both in the nucleus and membrane”. While GFP expression in root cell nuclei is observed, “membrane expression” may instead be a result of GFP in cytosolic space compressed against the membrane by the central cell vacuole. Without more detailed localization analysis, the authors should generalize this description.*

We have edited the text to state that GFP was only observed in mature tissues and not in the meristem. We have also included 3D reconstruction images of sections of the mature region (Supplementary Figure 9).

3. *Lines 213-230: The authors mention Klebsiella in preceding and later sections, but in these paragraphs the authors only characterize pC-HSL production by Pseudomonas and not Klebsiella. The authors should perform similar experiments to Sl Figure 17 with Klebsiella to better understand its pC-HSL production titers and ability to induce ‘receiver’ responses, relative to Pseudomonas.*

We have performed a similar experiment to the one done for *P. putida* and have reported the results in Supplementary Figure 20. We found that *K. pneumoniae* produced 300 ± 70 nM pC-HSL in the presence of *A. thaliana* pC-HSL receiver.

4. *Lines 300-304: The authors should state in the main text what concentration of arsenic was added in the plant environment.*

We have edited the main text as requested.

5. *Figure 1a: Image of root in maturation zone is upside down. Root hairs form on the root-ward side of epidermal cells and it is clear that this image was inadvertently flipped.*

The image has been rotated as suggested.

6. *Figure 2e: “Weight” is spelled incorrectly in the y-axis of two graphs.*

The typo has been corrected on both graphs.

7. *Figure 3a/3b: The authors should mention Klebsiella in the Figure 3a cartoon and in Figure 3b should show a picture of the Arabidopsis-containing Klebsiella.*

Figures 3a and 3b have been edited as suggested.

8. Figure 4b: The cartoon incorrectly shows IPTG/aTc inducing the arsenic sensor.

The typo has been corrected.

Reviewer #2:

1. *It was good to see that the response was still detectable in non-sterile soil, although it was much weaker than in sterile soil. In general, I think a word of caution should be added to the discussion about this. I am convinced that these reporter systems work well in closed systems, but once there is competition with other bacteria, possible signal breakdown or variable soil conditions in the field, the reliability of the sensors would be reduced. I am also not really convinced about examples of using this system to detect land mines. I know it's just a suggestion, but in reality, how would one first plant transgenic seeds (considering GM legislation) across large areas of potential landmine contaminated land, then harvest each plant, record the exact location, look for quantitative changes in fluorescence and then go back to the spot and decontaminate the mine. Yes, possible, maybe with drones, but not very practical.*

We have edited the discussion to add the caveat regarding performance in sterile versus non-sterile soil. Note that we are not the first to suggest or engineer the use of plants/microbes to detect landmines and, in fact, this is being tested in the field (not by us). Thus, we have kept the description in the discussion.

2. *Line 77: check spelling of Azorhizobium caulidurans*

We have corrected the spelling.

3. *Line 135: please correct to 'leading to GFP...'*

This line has been edited.

4. *Line 144: please correct this sentence: ..'allowed the plant roots to be exposed the plant roots to a homogeneous..'*

This sentence has been corrected.

5. *Line 160: In mature tissues GFP expression was both in the nucleus and the membrane': First, please refer here to GFP localization, not expression, Second, it is highly unlikely that GFP would localize in the membrane. This looks like cytoplasmic localization. The cytoplasm of these cells would be a thin strip pressed against the membrane.*

We have edited the text to describe it as GFP localization.

6. *Line 161: There was no GFP expression in the root hairs. Do you have an explanation for this observation?*

We have added Supplementary Figure 9 and videos that show GFP in the root hair cells when a nucleus was present.

7. *Line 163: ‘..showed induction of tissue throughout the root system’. Please rephrase to ‘showed induction of GFP expression throughout different root tissues..’ or similar.*

We have rephrased this statement as suggested.

8. *Line 365: please change ‘cinnomoyl’ to ‘cinnamoyl’*

This spelling error has been corrected.

9. *Line 465: Why was the UV laser used for GFP visualisation on the SP8? At ~365 nm excitation?*

Excitation for GFP was 488 nm on the Leica SP8. We have included this information in the methods. The terms ‘UV laser’ and ‘Argon laser’ which is the nomenclature of our instrument have been removed.

9. *Line 466: Please also give excitation wavelength of the argon laser.*

The excitation wavelength of the argon laser was 534 nm. The methods have been edited to include this information.

10. *For all figures with histograms, please show statistical differences in the figures, not just the legends.*

We have edited both the main text figures and the supplementary figures to include the statistical differences in the bar graphs.

11. *Figure 2: Thank you for adding phenotyping results for Arabidopsis. Did you record any root phenotypes since the highest expression of the reporter was in the root system?*

We performed a new experiment to compare the root phenotypes of the wild-type *A. thaliana* and the pC-HSL receiver. We found no significant differences between the two for both the primary root length and lateral root density (Figure 1f).

Reviewer #3:

1. *A. thaliana* lines containing the pC-HSL and OC12-HSL receivers showed 40-fold and 6-fold inductions, respectively (Figure 2a-c and Supplementary Figure 3). For the OHC14-HSL receiver, we only found a line that produced a low 2-fold dynamic range (Supplementary Figure 4) and the OC6-HSL receiver yielded no functional lines, so neither were pursued further. I can't find the information about the fold changes in the manuscript.

The information can be found in Figure 2b for the pC-HSL receiver, in Supplementary Figure 3 for the OC12-HSL receiver and in Supplementary Figure 4 for the OHC14-HSL receiver.

2. *In mature tissues, GFP expression was both in the nucleus and in the membrane. This information is not in the manuscript.*

We have removed this claim and have added images to Supplementary Figures 8 and 9.

3. *There was no GFP expression in the root hairs (Figure 2a, Supplementary Figures 6-7). The quality of the pictures and the level of details is not enough to conclude this.*

We have added Supplementary Figure 9 to show the root hair of the mature root region in more detail. We observed GFP in the root hair cells when a nucleus was present. We have edited the main text to include this information.

4. *No phenotypic differences were observed between wild-type *A. thaliana* and *A. thaliana* pC-HSL receiver (Figure 2b and Figure 2e). Knowing that the expression of the reporter is root specific, no phenotypes related with root architecture were quantified such as LR density, primary root length, etc.*

As suggested by the reviewer, we have compared the root phenotypes between the wild-type *A. thaliana* and the *A. thaliana* pC-HSL receiver. The results are provided in Figure 1f. No significant differences were found for either the primary root length or the lateral root density.

5. *No quantitative data is provided in Supl fig 15 and 16.*

We have now included statistical significance information on the bar graph of Supplementary Figure 15 (now Supplementary Figure 17). The quantification of fluorescence intensity of the images in Supplementary Figure 16 (now Supplementary Figure 18) are now shown with statistical significance information in Figure 3c.

6. *From these data, the concentration produced by the *P. putida* sender was estimated to be 70 ± 10 nM pC-HSL. This data is not in the manuscript.*

These data are in Supplementary Figure 19.

7. *The co-culture was grown for 24 hours and then the roots imaged. Plants grown for this length of time with bacteria did not show observable phenotypic changes as compared to the plants grown without the bacteria (Figure 3b). Phenotypes were not quantified.*

We have deleted this claim.

8. *Uninduced when grown with wild-type bacteria (Figures 3c-d, Supplementary Figures 15-16, 19). The presence of 100 μ M *p*-coumarate in the medium increased the response of the receiver to *P. putida* pTT337 from 11- to 15-fold (Supplementary Figure 20). Where I can see this induction?*

We have deleted this claim.

9. *The new soil experiments are open making impossible the establishment of causality between the bacterium adding and the plant signal.*

The soil experiments are closed, at least in how we understand the definition, and described in the methods. There should not be a problem inferring causality from these experiments.

Reviewers' Comments:

Reviewer #1:

Remarks to the Author:

The authors have addressed all my remaining concerns. The work represents an outstanding advance in engineering transkingdom associations between plants and rhizobacteria.

Reviewer #2:

Remarks to the Author:

Thank you for addressing all the comments with additional data, figures and statistical analysis. All my queries are resolved in the revised manuscript.

Reviewer #3:

None